# A Semantically Consistent Dataset for Data-Efficient Query-Based Universal Sound Separation

Kai Li [* 1 2]   Jintao Cheng [* 1]   Chang Zeng [3]   Zijun Yan [1]   Helin Wang [4]   Zixiong Su [3]   Bo Zheng [3]   Xiaolin Hu [1 2 5]

## Abstract

Query-based universal sound separation is fundamental to intelligent auditory systems, aiming to isolate specific sources from mixtures. Despite recent advances, existing methods continue to suffer from residual interference in complex acoustic scenes. This performance limitation stems largely from a data bottleneck: in-the-wild datasets contain weak labels and severe co-occurrence of events. These flaws induce models to learn spurious correlations between background noise and target categories instead of robust acoustic features. To address this, we propose an automated pipeline that eliminates co-occurrence of events by mining high-purity single-event segments from in-the-wild datasets via a semantically consistent synthesis protocol. Utilizing this pipeline, we constructed *Hive*, a high-quality synthetic dataset comprising 2.4k hours of raw audio. Experimental results demonstrate that, compared with the state-of-the-art model SAM-Audio which was trained on a huge dataset $\sim$500 times larger than Hive, certain open-source models trained on Hive achieve competitive separation accuracy and perceptual quality. Moreover, these models exhibited remarkable zero-shot generalization on out-of-distribution evaluation benchmarks. These findings highlight that prioritizing purity of supervised signals enables significant data efficiency, offering a new paradigm for training robust auditory foundation models with reduced computational costs. Code and dataset are available at https://cslikai.cn/Hive.

---
*Equal contribution [1]Department of Computer Science and Technology, Institute for AI, BNRist, Tsinghua University, Beijing, China. [2]IDG/McGovern Institute for Brain Research, Tsinghua University, Beijing, China. [3]Shanda AI Research Tokyo. [4]Johns Hopkins University. [5]Chinese Institute for Brain Research (CIBR), Beijing, China. Correspondence to: Xiaolin Hu <xlhu@tsinghua.edu.cn>.

*Proceedings of the 43$^{rd}$ International Conference on Machine Learning*, Seoul, South Korea. PMLR 306, 2026. Copyright 2026 by the author(s).

## 1. Introduction

Real-world acoustic environments are inherently polyphonic and contain multiple overlapping sound events (Heittola et al., 2013). While the human auditory system effectively isolates target sources from mixtures, a capability known as the "Cocktail Party Effect" (Cherry, 1953; Arons, 1992), previous computational methods have focused on restricted domains like speech (Li et al., 2023; 2025) or music (Uhlich et al., 2024). Recently, computational auditory scene analysis (CASA) has expanded toward query-based universal sound separation (USS) (Lee et al., 2025a; Wang et al., 2025). In contrast to domain-specific methods, query-based USS targets arbitrary sound categories, including environmental and mechanical events. This capability is pivotal for applications such as immersive audio rendering (Gupta et al., 2022), machine hearing (Lyon, 2010), and intelligent audio editing (Yan et al., 2025).

Conventional blind source separation (Li et al., 2023) is constrained by fixed output cardinality and the permutation problem, limiting its applicability to open-domain scenarios (Li et al., 2025). To address these limitations, query-based universal sound separation has emerged, leveraging multimodal prompts (e.g., text, audio, visual) to explicitly target specific sources (Kavalerov et al., 2019). Early approaches adopted discriminative paradigms, ranging from end-to-end architectures (Liu et al., 2022) to weakly supervised visual bridging (Dong et al., 2023) and large-scale open-domain frameworks (Liu et al., 2024). Recently, the field has shifted toward generative paradigms, such as FlowSep utilizing flow matching (Yuan et al., 2025), while DGMO (Lee et al., 2025b), ZeroSep (Huang et al., 2025) and ZETA (Manor & Michaeli, 2024) explore unsupervised strategies via test-time optimization or zero-shot settings. More recently, research has advanced toward unified prompting, where frameworks like OmniSep (Cheng et al., 2025) and SAM-Audio (Shi et al., 2025) integrate diverse modalities, including text, vision, and time segments, aiming to enhance controllability and facilitate separation-on-demand.

Despite architectural evolution across both discriminative and generative paradigms, query-based USS methods persistently exhibit residual interference, characterized by the audible leakage of background noise or concurrent events

*Table 1.* Overview of the original dataset and mixing pipelines adopted by USS methods. "Train. Dur." and "Val. & Test Dur." refer to the total duration of the original unmixed training set and the combined validation and test sets, respectively. "Principled mix": whether the mixing process follows explicit semantic or acoustic constraints rather than random combinations. "Pub. mixed data and Pub. mix tool": whether pre-mixed data and mixing code are publicly released. (✓= yes, ◐= partially, ✗= no)

| Dataset | Train. Dur. (h) | Val. & Test Dur. (h) | Sample-rate | Single-label | Principled mix | Pub. mixed data | Pub. mix tool |
|---|---|---|---|---|---|---|---|
| (Liu et al., 2022) | 17.3 | ~4 | 32k | ✗ | ✗ | ✗ | ✗ |
| (Dong et al., 2023) | 550 | ~50 | 16k | ✗ | ✗ | ✗ | ✗ |
| (Liu et al., 2024) | 14,100 | ~109 | 32k | ◐ | ✗ | ✗ | ✓ |
| (Yuan et al., 2025) | 1,680 | ~9 | 16k | ◐ | ✗ | ✗ | ✗ |
| (Cheng et al., 2025) | 560 | ~10 | 16k | ◐ | ✗ | ✗ | ✗ |
| (Shi et al., 2025) | ~1,000,000 | ~47 | 48k | ◐ | ✓ | ✗ | ✗ |
| Hive | 2,442 | 292 | 44.1k | ✓ | ✓ | ✓ | ✓ |

into the separated output (Huang et al., 2024; Hai et al., 2024; Lee et al., 2025b). We attribute this deficiency to systematic biases in training data. Current training paradigms predominantly rely on large-scale in-the-wild datasets, such as AudioSet (Gemmeke et al., 2017) and VGGSound (Chen et al., 2020), due to their vast scale and category diversity. However, these datasets inevitably suffer from weak labels and high co-occurrence of events, resulting in severe label-signal misalignment (Fonseca et al., 2021). Consider the case where "Rain" clips frequently contain concurrent wind or traffic: lacking fine-grained supervision, models inevitably hallucinate these environmental artifacts as inherent acoustic characteristics of the target category. This systematic bias causes the model to treat background noise as part of the desired signal. Given these constraints, the scarcity of high-purity, single-event supervision remains a critical bottleneck in training robust USS models (Kong et al., 2023; Wisdom et al., 2021).

Prevailing paradigms prioritize scaling both dataset size and model capacity, exemplified by unified models trained on millions of audio hours (Shi et al., 2025). While yielding impressive performance, such brute-force scaling imposes prohibitive computational costs, creating significant barriers to reproducibility and accessibility for the broader research community. It motivates us to ask: *Is it possible to achieve competitive separation performance with significantly greater data efficiency?*

To answer this question, we propose an automated pipeline for high-quality data cleaning and synthesis in query-based USS. It mines high-purity single-event segments from complex auditory scenes via a unified label system and rigorous quality control. We then propose a semantically consistent strategy to mix the single-event segments, yielding a new dataset, named *Hive*. It comprises ~2.4k hours of high-quality raw audio. We trained representative discriminative (AudioSep) and generative (FlowSep) models on Hive and evaluated them alongside million-hour scale baselines like SAM-Audio. Remarkably, despite utilizing only $\sim 0.2\%$ of the data scale used by SAM-Audio, our trained models achieved competitive perceptual quality to SAM-Audio. These results empirically indicate that prioritizing purity

of supervised signals offers a data-efficient alternative to brute-force scaling, providing a reproducible pathway for the efficient training of query-based USS models.

**Conflict of Interest Disclosure.** One of the authors of this paper (H. Wang) is also a co-author of SAM-Audio (Shi et al., 2025), which serves as the primary million-hour-scale baseline benchmarked in this work. All comparisons against SAM-Audio reported here are conducted exclusively through its publicly released model checkpoint and inference code, without privileged access to its training data, internal logs, or any unreleased variant; the empirical setup is otherwise identical to that applied to all other baselines. Beyond this co-authorship, the authors declare no further competing financial interests. Per ICML policy, the authors' academic and industrial affiliations (Tsinghua University, Johns Hopkins University, and Shanda AI Research Tokyo) do not by themselves constitute conflicts of interest; in particular, no proprietary model, dataset, or service of Shanda AI Research Tokyo was used, evaluated, or compared against in this paper, and the work received no targeted funding from any of these affiliations beyond standard institutional support.

## 2. Related Work

### 2.1. Query-Based Universal Sound Separation Methods

Blind Source Separation (BSS) has traditionally relied on internal statistics but is often constrained by the permutation problem and fixed source counts (Liutkus et al., 2013). To address these issues, the field has shifted toward query-based USS, which utilizes auxiliary prompts (e.g., text, audio) to isolate arbitrary targets from complex environments (Li et al., 2025). Current USS approaches are broadly categorized into discriminative and generative paradigms.

Discriminative paradigm treats separation as a signal estimation task, learning a direct mapping function to filter the target from the mixture, typically via time-frequency masking. LASS-Net (Liu et al., 2022) pioneered this by introducing language queries, while AudioSep (Liu et al., 2024) and CLIPSep (Dong et al., 2023) further scaled this paradigm

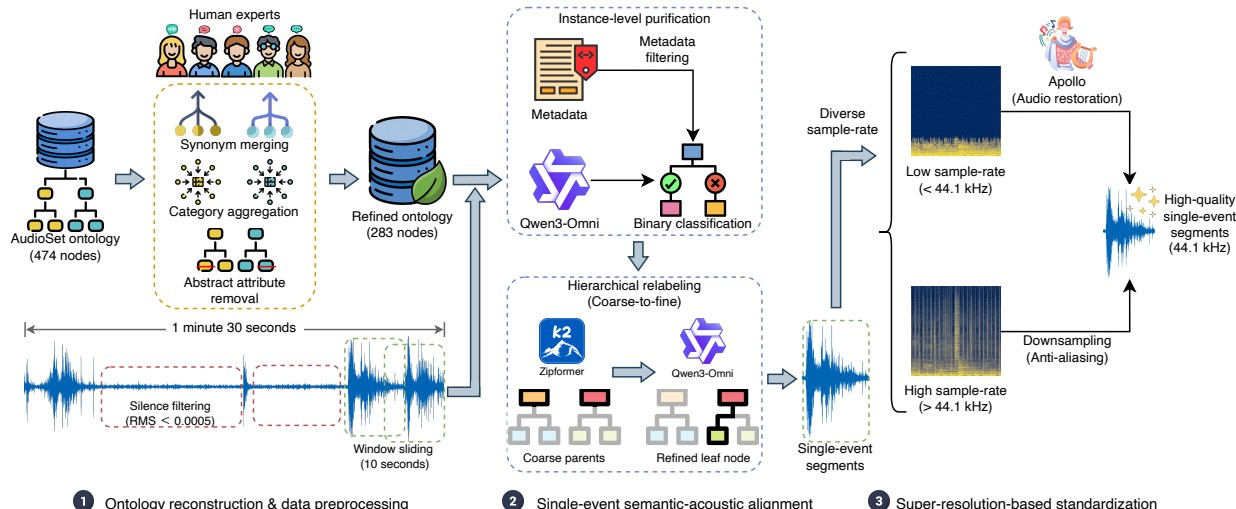

*Figure 1.* Overview of the proposed pipeline. The framework consists of three coupled stages: (1) ontology reconstruction & data preprocessing. (2) single-event semantic-acoustic alignment. (3) super-resolution-based standardization.

using contrastive audio-text or image-text alignment. Although computationally efficient, these methods are often constrained by the fixed resolution of the input representation. Conversely, generative paradigm models the data distribution of the source signal, formulating separation as conditional synthesis to reconstruct targets from noise or latent representations. This category includes FlowSep (Yuan et al., 2025), which utilizes flow matching, and training-free methods like DGMO (Lee et al., 2025b) and ZETA (Manor & Michaeli, 2024) that leverage pretrained priors. Recently, unified frameworks such as OmniSep (Cheng et al., 2025) and SAM-Audio (Shi et al., 2025) have integrated these capabilities with multimodal prompts. In a parallel line of work, target speech and target sound extraction further broaden this conditioning interface beyond category-level queries, exploring enrollment-, text-, and audio-conditioned source extraction (Li et al., 2025; Hai et al., 2024; Wang et al., 2025).

However, despite these architectural advances, models trained on weakly labeled in-the-wild data continue to suffer from severe interference in complex scenarios. We argue that this bottleneck stems from fundamental label-signal misalignment in current training datasets, suggesting that further progress hinges on the construction of high-quality, event-aligned data rather than scale or architecture alone.

### 2.2. Data Cleaning and Synthesis

Current USS methods rely heavily on data synthesis using in-the-wild datasets like AudioSet (Gemmeke et al., 2017) and VGGSound (Chen et al., 2020) (see Table 1). Although extensive, these datasets are inherently polyphonic and suffer from severe co-occurrence of events (e.g., wind inherent in rain recordings). This introduces systematic supervision noise, as models inevitably learn to associate background

artifacts with the target category, establishing a performance ceiling on separation purity (Fonseca et al., 2021). Furthermore, training data is typically constructed via naive random mixing, which ignores semantic consistency (Liu et al., 2022). Even on the source side, existing data resources sidestep rather than solve segment-level purification: Scaper provides flexible event-level mixing but presupposes a curated bank of isolated recordings (Salamon et al., 2017), FUSS releases controlled mixtures without addressing upstream source purification (Wisdom et al., 2021), and large-scale weakly labeled efforts such as GASS scale up training mixtures while inheriting impurity from their in-the-wild sources (Pons et al., 2024). While some methods attempt to filter anchors using pre-trained event detectors (Shi et al., 2025; Wisdom et al., 2021), these detectors are often trained on the same noisy data, leading to a circular propagation of bias. An orthogonal line of work, exemplified by FlexSED, leverages LLM-derived semantic knowledge for data quality but targets negative query filtering in open-vocabulary sound event detection rather than USS mixture construction (Hai et al., 2025). Consequently, achieving robust separation without brute-force scaling requires a dual solution: mining high-purity single-event segments to fix the source, and implementing semantically consistent synthesis to fix the process (Kong et al., 2023).

## 3. Single-Event Data Collection Pipeline

In-the-wild audio recordings are widely utilized in auditory research due to their massive scale and acoustic diversity (Liu et al., 2024; Shi et al., 2025), serving as a cornerstone for generalizing to real-world environments. However, these uncurated data typically suffer from complex acoustic backgrounds and high co-occurrence of events, rendering raw supervision signals unreliable for high-fidelity separation

tasks (Fonseca et al., 2021). To mitigate this, we propose an automated pipeline designed to mine high-quality single-event segments from uncurated corpora and assign precise semantic annotations. As illustrated in Figure 1, the pipeline operates through three distinct stages: ontology reconstruction & data preprocessing, single-event semantic-acoustic alignment, and super-resolution-based standardization.

## 3.1. Ontology Reconstruction and Data Preprocessing

A discriminative and semantic-consistent label system is fundamental for USS. We adopted the AudioSet ontology (474 leaf nodes) as a foundational taxonomy (Gemmeke et al., 2017). However, AudioSet exhibits significant semantic overlap and excessive granularity, which impedes effective class separation. To address this, we execute a rigorous taxonomy reconstruction process validated by human experts. First, we merged synonymous or highly overlapping labels to reduce ambiguity. For instance, action-oriented labels (e.g., "Drum beat") are unified with their corresponding entities ("Drum"). Second, fine-grained categories with minimal acoustic distinctiveness like specific biological sounds ("Fowl" and "Coo") were aggregated into their superordinate concepts. Furthermore, we explicitly excluded labels describing broad acoustic environments (e.g., "indoor", "countryside"), format tags (e.g., "MP4"), and abstract attributes. These categories typically represent background textures rather than separable, localizable foreground events. This pruning yielded a refined, separation-oriented label space of 283 leaf nodes (see Table A1). The detailed reconstruction protocol is provided in Appendix A.

Following the ontology definition, we standardized raw audio streams into uniform tensors. We employed a sliding window strategy with a 10s duration and a 5s overlap. To filter invalid silence, segments with root mean square (RMS) energy below $5 \times 10^{-4}$ were automatically discarded (Uhlich et al., 2024). Finally, to ensure uniform sequence length across the dataset and avoid padding-induced artifacts, any residual tail segment shorter than the full 10s window duration was discarded.

## 3.2. Single-Event Semantic-Acoustic Alignment

The primary challenge in constructing separation datasets lies in resolving the ambiguity between acoustic co-occurrence of events and weak semantic labels (Fonseca et al., 2021; Kong et al., 2023). To address this, we propose a cascaded alignment framework designed to ensure samples exhibit acoustic singularity while mapping precisely to mutually exclusive ontology leaf nodes.

For instance-level purification, we maximized purity by filtering samples based on metadata, explicitly excluding segments tagged with multiple labels. However, to tackle unannotated background noise or transient interference, we intro-

duced a content-based neural polyphony detector. Leveraging the multimodal capabilities of Qwen3-Omni (Xu et al., 2025), we performed zero-shot binary classification to reject segments containing multi-event mixtures, thereby guaranteeing the acoustic isolation of training samples.

While this establishes event singularity, precise semantic alignment under long-tail distributions remains non-trivial. We adopted a coarse-to-fine strategy leveraging ontology topology. First, an audio-tag model[1] predicted coarse-grained parent nodes, pruning ambiguous samples via a confidence threshold of 0.7 (Yao et al., 2024). Subsequently, using the predicted coarse category as a semantic prior, we employed Qwen3-Omni to refine the classification into specific leaf nodes within the restricted candidate subset. Samples failing to match any candidate leaf nodes are rejected. This design effectively integrates discriminative robustness with generative reasoning, ensuring precise alignment from acoustic signals to ontology labels. The detailed prompts employed by Qwen3-Omni are provided in Appendix B.

## 3.3. Super-Resolution-Based Standardization

In-the-wild datasets exhibit significant sampling rate heterogeneity due to diverse acquisition conditions. Naive integration of such data introduces unnatural spectral discontinuities, biasing models to ignore high-frequency details. To ensure spectral consistency, we standardized the global sampling rate to 44.1 kHz via a dual-strategy approach. For bandwidth-limited segments, we leveraged the Apollo model to plausibly reconstruct high-frequency harmonics from low-frequency priors (Li & Luo, 2025). Conversely, high-resolution inputs exceeding the target rate were downsampled via anti-aliasing filters, eliminating computational redundancy while preserving spectral integrity.

## 3.4. Source Curation and Properties

We aggregated raw audio from 12 different public datasets, including AudioSet (Gemmeke et al., 2017), VGGSound (Chen et al., 2020), FreeSound (Fonseca et al., 2017), and BBC Sound Effects (British Broadcasting Corporation, 1991). Unlike controlled studio recordings, these in-the-wild sources capture complex acoustic environments and long-tail event distributions, which are essential for domain generalization. Following the rigorous cleaning and alignment as described in Section 3.1 and 3.2, the final source pool comprises ~0.9M unique clips totaling ~2,442 hours. As shown in Figure 2, the distribution exhibits a realistic long-tail structure: general-domain datasets (BBC, AudioSet) constitute the backbone ($> 77\%$), while specialized subsets (e.g., DCASE (Mesaros et al., 2019), ESC50 (Piczak, 2015)) supplement rare events to prevent domain overfitting.

---

[1] https://k2-fsa.github.io/sherpa/onnx/audio-tagging/index.html

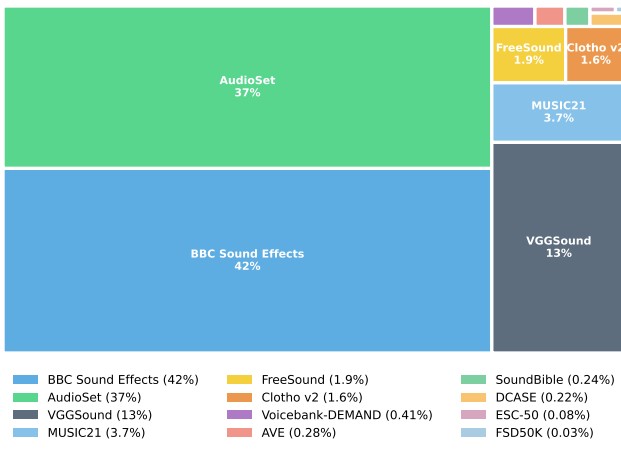

*Figure 2.* Proportional composition of the 12 heterogeneous source datasets.

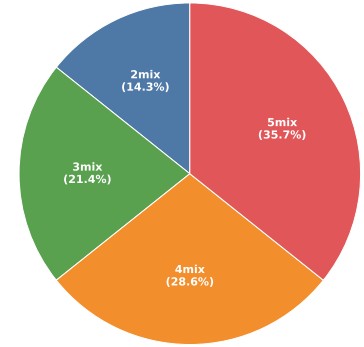

*Figure 3.* Distribution of the number of sources in the Hive.

Detailed compositions are provided in Appendix C. The pipeline runs as a one-time offline process and completes in under 90 GPU-hours on $4\times$ RTX 3090s.

## 4. Dataset Construction

### 4.1. Semantically Consistent Mixing Strategy

Given the high-quality single-event segments obtained from the proposed pipeline, we proceed to synthesize training mixtures. However, naive random mixing often yields implausible combinations (e.g., aquatic animals co-occurring with urban traffic), introducing incorrect contextual priors. To mitigate this, we propose a semantic compatibility protocol governed by a logical co-occurrence matrix.

We define a binary matrix $\mathbf{M} \in \{0, 1\}^{N \times N}$ where entries indicate semantic validity. $\mathbf{M}$ is derived via Qwen3-Omni using the prompt in Figure A3, which enforces strict criteria: events must be capable of coexisting without spatiotemporal conflict (e.g., allowing "typing" with "air conditioning" while rejecting logical contradictions); a representative submatrix is shown in Figure 5. Based on $\mathbf{M}$, we generated training data. For each mixture, a source count $C$ is sampled from $\{2, \ldots, 5\}$. We construct each tuple by sampling an anchor event and iteratively adding sources from categories compatible with all already selected events, ensuring pair-

wise compatibility ($\mathbf{M}_{l_i, l_j} = 1, \forall i, j$). Valid tuples undergo duration normalization (4s training, 10s testing) and energy unification (RMS $= 0.1$). The mixture $\mathbf{y}$ follows an additive superposition model: $\mathbf{y} = \mathbf{x}_1 + \sum_{c=2}^{\check{C}} 10^{\text{SNR}_c/20} \cdot \mathbf{x}_c$, where $\mathbf{x}_c$ denotes the $c$-th normalized source. Relative gains are determined by SNRs randomly sampled from $[-5, 5]$ dB, simulating diverse acoustic conditions and enabling the model to adapt to varied energy ratios.

### 4.2. Hive Dataset Statistics

Hive serves as a large-scale dataset comprising 19.6 million mixtures ($\sim$22.4k hours). The dataset is partitioned into training (17.5M), validation (1.75M), and test (350k) sets. Notably, as detailed in Table 1, Hive reserves 292 hours of distinct unmixed sources for validation and testing. This scale significantly exceeds the evaluation sets of prior large-scale baselines (e.g., 109 hours for AudioSep and 47 hours for SAM-Audio), ensuring that our zero-shot evaluation covers a far broader spectrum of unseen acoustic events. While training and validation samples are cropped to 4 seconds for computational efficiency, test samples utilize 10-second clips to rigorously evaluate temporal consistency and long-term dependency modeling. Crucially, Hive adopts a complexity-biased distribution rather than a uniform one. As shown in Figure 3, dense scenarios with 5 concurrent sources dominate the training set ($\sim 35\%$), effectively pushing the boundaries of separation robustness. The synthesized mixtures maintain broad coverage across the 283-class ontology. Figure 4 provides an overview of the label frequency distribution, while a complete per-class breakdown over all 283 labels is provided in Appendix E.

## 5. Experiments Settings

**Datasets and Benchmarks** We evaluated zero-shot generalization and robustness by testing the original pretrained checkpoints of all baselines on the Hive test set without fine-tuning. This setting highlights the difficulty of semantically consistent but acoustically dense scenes in Hive. To assess Hive as a training resource, we trained AudioSep and FlowSep from scratch on Hive, and compared them with their original checkpoints on three out-of-domain benchmarks: MUSDB18-HQ (Rafii et al., 2019), USS-Bench[2], and VGGClean_eval, the public VGGSound-based separation benchmark released by AudioSep (Liu et al., 2024). See Appendix F for details.

**Baselines and Metrics** We compared discriminative models (LASS-Net (Liu et al., 2022), CLIPSep (Dong et al., 2023), AudioSep (Liu et al., 2024), OmniSep (Cheng et al., 2025)) and generative models (ZETA (Manor & Michaeli, 2024), FlowSep (Yuan et al., 2025), ZeroSep (Huang et al.,

---

[2]https://mab.to/GsSVrdZK0dfQ1/us3

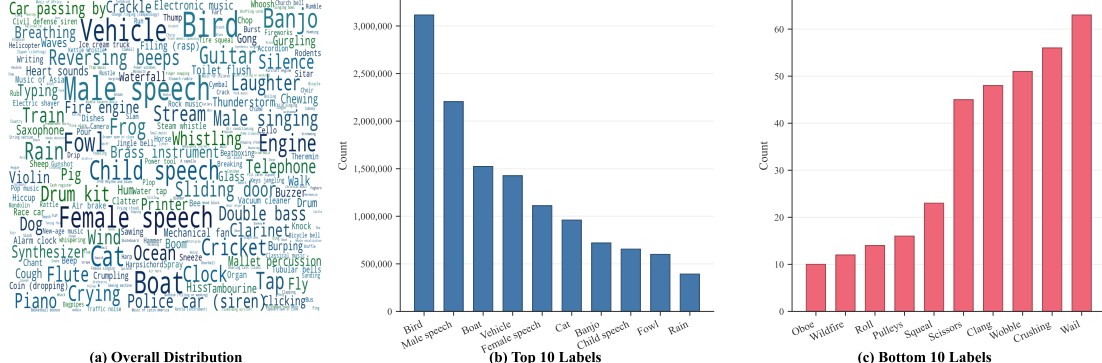

*Figure 4.* Label frequency statistics of Hive dataset. (a) Overall label distribution visualized as a word cloud (token size ∝ mixture count). (b) Top-10 most frequent labels. (c) Bottom-10 least frequent labels.

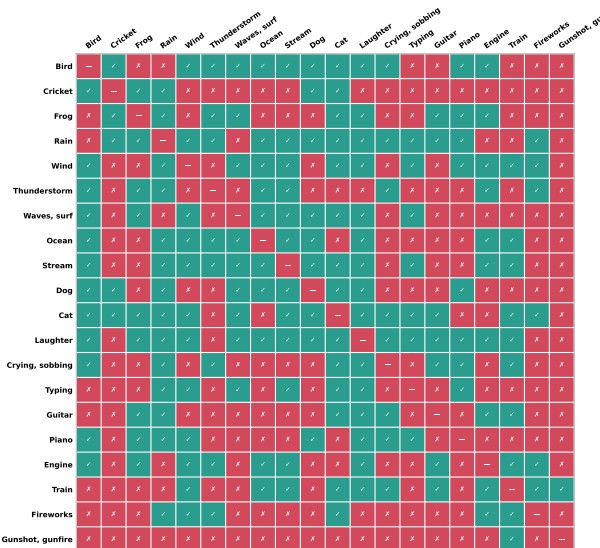

*Figure 5.* Representative submatrix of the semantic compatibility matrix used in Hive. The matrix guides whether two event categories can naturally co-occur, supporting mixture construction that avoids semantically implausible event combinations.

2025), DGMO (Lee et al., 2025b), SAM-Audio (Shi et al., 2025)). Performance is quantified across three dimensions: (1) *Signal Fidelity*: SDR (Vincent et al., 2006) and SI-SDR (Le Roux et al., 2019); (2) *Perceptual & Semantic Quality*: FAD (Gui et al., 2024), LPAPS (Manor & Michaeli, 2024), CLAP similarity (CLAP Audio, CLAP Text) (Elizalde et al., 2023), and MUSHRA listening scores; (3) *Reference-free Quality*: SAM-Audio Judge (Shi et al., 2025), reporting overall quality (OQ), recall (Rec), precision (Pre), and faithfulness (Fai) as secondary automated evidence. Detailed configurations for baselines and metrics are provided in Appendix G.

**Implementation Details** All experiments were implemented in PyTorch and conducted on $8\times$ NVIDIA A100 (80GB) GPUs. To ensure fair comparison, we adhered to the original hyperparameter architectures of the trained models. AudioSep was trained with a batch size of 64 using

*Table 2.* Comparison on the 4-AFC audio event recognition task ($n = 100$ clips, chance = 25%). Human Average reports per-participant accuracy against the consensus label obtained by majority voting over 67 valid participants; LALM rows report model-vs-consensus accuracy. Best result in **bold**.

| Method | Accuracy (%) |
|---|---|
| Human Average | 90.75 |
| Gemini 3.1 Pro | 95.00 |
| GPT-Audio | 90.00 |
| **Qwen3-Omni (Ours)** | **98.00** |

*Table 3.* Effect of the semantic-consistency constraint on mixture construction. Both regimes used 175k Hive mixtures and identical training pipelines. Within each model, the better strategy is in **bold**.

| Strategy | Model | SDR↑ | SI-SDR↑ | LPAPS↓ | CLAP-T↑ | OQ↑ | Pre↑ |
|---|---|---|---|---|---|---|---|
| Consistency-guided | AudioSep | **4.12** | **3.37** | **4.10** | **0.29** | **3.11** | **3.16** |
| Random mixing | AudioSep | 3.12 | 2.35 | 4.19 | 0.24 | 2.96 | 3.02 |
| Consistency-guided | FlowSep | – | – | **4.24** | **0.17** | **2.79** | **3.01** |
| Random mixing | FlowSep | – | – | 4.35 | 0.13 | 2.64 | 2.88 |

AdamW (Loshchilov & Hutter, 2017) and a learning rate of $1 \times 10^{-3}$, decayed by a factor of 0.5 upon validation plateaus (patience=20). FlowSep utilized a constant learning rate of $5 \times 10^{-5}$. Both models were trained for ~3M steps to ensure convergence. During evaluation, all outputs were resampled to 44.1 kHz.

## 6. Results

### 6.1. Reliability of LLM-based Relabeling

To quantitatively validate the label purity achieved by our pipeline, we conducted a 4-alternative forced choice (4-AFC) identification task on 100 clips. Each trial presented a 10-second audio clip and four same-level candidate labels, and the consensus label was determined by majority voting among 67 valid human participants. Inter-rater reliability was high (Fleiss' $\kappa = 0.843$ (Fleiss, 1971), indicating almost perfect agreement on the scale of Landis & Koch (1977)), and individual annotators agreed with the consensus on 90.75% of clips on average. We further compared

*Table 4.* Zero-shot results on the Hive test set (averaged 2-to-5 mix). Models marked with "(Hive)" were trained on the Hive dataset. Detailed results are in Table A3. Best results are **bold**, second-best are wavy underlined. Generative models without waveform preservation guarantees are marked with "–" for signal fidelity metrics; unavailable MUSHRA scores are also marked with "–".

| Model | Signal Fidelity ↑ | | Perceptual & Semantic Quality | | | | | Reference-free Quality ↑ | | | |
|---|---|---|---|---|---|---|---|---|---|---|---|
| | SDR | SI-SDR | LPAPS ↓ | FAD ↓ | CLAP-A ↑ | CLAP-T ↑ | MUSHRA ↑ | OQ | Rec | Pre | Fai |
| *Discriminative Models* | | | | | | | | | | | |
| LASS-Net (Liu et al., 2022) | -2.97 | -4.04 | 4.78 | 1.04 | 0.43 | 0.14 | 39.4 | 2.44 | 4.42 | 2.48 | 4.06 |
| CLIPSep (Dong et al., 2023) | -6.20 | -9.38 | 5.95 | 1.05 | 0.43 | 0.10 | 30.7 | 2.45 | 4.11 | 2.63 | 3.37 |
| AudioSep (Liu et al., 2024) | 2.37 | 1.58 | 4.22 | 0.90 | 0.57 | 0.26 | 60.9 | 2.94 | **4.86** | 2.96 | **4.47** |
| OmniSep (Cheng et al., 2025) | -2.85 | -4.67 | 4.93 | 0.96 | 0.52 | 0.16 | 45.1 | 2.65 | 4.58 | 2.76 | 3.82 |
| AudioSep (Hive) | **5.67** | **5.02** | **3.86** | **0.80** | **0.65** | **0.31** | **68.4** | **3.34** | 4.81 | **3.40** | 4.39 |
| *Generative Models* | | | | | | | | | | | |
| ZETA (Manor & Michaeli, 2024) | – | – | 5.43 | 1.04 | 0.44 | 0.28 | – | 2.82 | 4.10 | **3.54** | 3.19 |
| FlowSep (Yuan et al., 2025) | – | – | 4.18 | 0.87 | 0.57 | 0.16 | 54.7 | 2.76 | 4.06 | 2.90 | 3.69 |
| ZeroSep (Huang et al., 2025) | – | – | 4.50 | 0.92 | 0.55 | 0.25 | 58.8 | 2.87 | 4.76 | 2.97 | 4.27 |
| DGMO (Lee et al., 2025b) | – | – | 4.94 | 0.97 | 0.51 | 0.27 | 53.6 | 2.85 | 4.48 | 3.19 | 3.85 |
| SAM-Audio (Shi et al., 2025) | – | – | 5.21 | 1.03 | 0.41 | 0.16 | 62.6 | 2.90 | 3.77 | 3.33 | 3.63 |
| FlowSep (Hive) | – | – | 4.25 | 0.84 | 0.61 | 0.19 | 61.8 | 3.05 | 4.18 | 3.35 | 3.70 |

Qwen3-Omni, Gemini 3.1 Pro, and GPT-Audio against the human consensus under the same multiple-choice protocol. As summarized in Table 2, Qwen3-Omni achieved 98.0% agreement with human consensus, while Gemini 3.1 Pro and GPT-Audio achieved 95.0% and 90.0%, respectively. These results support the reliability of the re-labeling and purification stage under this protocol. Class-frequency-stratified analysis showed that head classes generally achieved higher consensus, while more ambiguous middle- and tail-frequency classes, such as Drum kit, Guitar, and Rain, exhibited lower agreement.

## 6.2. Validation of Semantic Consistency

We further validated the semantic-consistency constraint introduced in Section 4, which restricted mixture construction to event combinations deemed semantically consistent by $\mathbf{M}$. To isolate its contribution from source purity, we trained AudioSep and FlowSep on 175k Hive mixtures under two regimes: (i) *Consistency-guided*, the default sampling that enforced $\mathbf{M}_{l_i,l_j} = 1$ for all pairs in a tuple; (ii) *Random mixing*, sampling tuples uniformly over Hive's purified single-event sources without the consistency constraint. All other factors, including the source pool, source count distribution, duration normalization, energy unification, and SNR sampling, were held fixed.

As shown in Table 3, random mixtures built from Hive's purified single-event sources already delivered strong performance, indicating that source purity alone removed a substantial fraction of supervision noise. Enforcing the semantic-consistency constraint yielded consistent additional gains across both architectures, including a 1.0 dB SDR improvement for AudioSep and improvements on perceptual and reference-free metrics for both models. These results indicated that source purity and semantic consistency acted as complementary (rather than redundant) axes of data

quality in Hive.

## 6.3. Zero-shot Benchmarking on the Hive Test Set

**Performance Analysis**. AudioSep achieves the strongest SDR (2.37 dB) among prior systems, while early methods such as CLIPSep collapse to negative SDR, consistent with a heavy reliance on co-occurrence shortcuts that Hive's decorrelated mixtures expose (Wisdom et al., 2021); we quantify this dependence directly in Section 6.5. Generative models such as SAM-Audio achieve competitive perceptual scores (FAD, LPAPS) but lag in semantic fidelity (CLAP-T), reflecting the tendency of conditional synthesis to drift on content while preserving naturalness. Performance also degrades consistently as mixtures scale from 2 to 5 sources (Table A3), indicating that acoustic density remains a critical bottleneck even with precise semantic alignment. Audio samples and spectrogram visualizations are available on our demo page[3]. The MUSHRA listening test[4] corroborates these gains with direct human judgments. AudioSep (Hive) attains a score of 68.4, ahead of SAM-Audio (62.6) and AudioSep (Orig.) at 60.9; FlowSep (Hive) improves over FlowSep (Orig.) by 7.1 points. The Hive gain is larger for AudioSep than for FlowSep, consistent with mask-based separation, which preserves the mixture phase and spectrum, benefiting more directly from high-purity single-event supervision than a noise-prior conditional synthesizer.

**Efficiency Analysis**. Table A4 reports parameters, MACs, latency, and memory for all baselines on 1-second audio. Discriminative models retain a sizeable efficiency margin: AudioSep runs at 0.02 s GPU time within ∼1.1 GB of memory, whereas SAM-Audio (8.2 B parameters, ∼32 GB) and iterative generative methods such as DGMO (∼4.9 × 10³ s

---

[3] https://cslikai.cn/Hive
[4] Scores on the standard 0–100 MUSHRA scale; protocol details in Appendix G.2.

*Table 5.* Performance comparison on third-party test sets. "Orig." stands for Original Datasets, and unreported metrics are marked with "–".

| Model (Train Set, Scale) | Signal Fidelity ↑ | | Perceptual & Semantic Quality | | | | Reference-free Quality ↑ | | | |
|---|---|---|---|---|---|---|---|---|---|---|
| | SDR | SI-SDR | LPAPS ↓ | FAD ↓ | CLAP-A ↑ | CLAP-T ↑ | OQ | Rec | Pre | Fai |
| *MUSDB18-HQ* | | | | | | | | | | |
| AudioSep (Orig., 14.1k h) | -1.01 | -1.46 | 4.94 | 0.97 | 0.49 | 0.09 | 2.39 | 3.92 | 2.40 | 3.84 |
| FlowSep (Orig., 1.7k h) | – | – | 4.72 | 0.94 | 0.54 | 0.11 | 2.21 | 3.54 | 2.30 | 3.33 |
| SAM-Audio (Orig., 1M h) | – | – | **4.33** | **0.83** | **0.65** | 0.12 | **3.24** | **4.54** | **3.68** | **4.27** |
| AudioSep (Hive, 2.4k h) | **1.36** | **0.89** | 4.68 | 0.91 | 0.58 | **0.15** | 3.01 | 4.18 | 3.14 | 4.05 |
| FlowSep (Hive, 2.4k h) | – | – | 4.34 | **0.83** | 0.65 | 0.12 | 3.11 | 4.22 | 3.22 | 3.90 |
| *USS-Bench* | | | | | | | | | | |
| AudioSep (Orig., 14.1k h) | -1.86 | -3.63 | 5.02 | 0.91 | 0.58 | 0.11 | 2.97 | 4.01 | 3.04 | **4.17** |
| FlowSep (Orig., 1.7k h) | – | – | 5.19 | 0.93 | 0.56 | 0.12 | 2.33 | 3.56 | 2.58 | 3.14 |
| SAM-Audio (Orig., 1M h) | – | – | 4.96 | 0.90 | 0.57 | 0.15 | 3.47 | 4.14 | **3.77** | 3.91 |
| AudioSep (Hive, 2.4k h) | **2.29** | **0.30** | **4.22** | **0.75** | **0.69** | **0.22** | **3.56** | **4.22** | 3.68 | 4.03 |
| FlowSep (Hive, 2.4k h) | – | – | 4.62 | 0.88 | 0.58 | 0.15 | 2.89 | 3.80 | 3.34 | 3.75 |
| *VGGClean_eval* | | | | | | | | | | |
| AudioSep (Orig., 14.1k h) | **8.67** | **7.48** | **0.58** | **0.60** | **0.79** | **0.25** | 3.42 | 4.85 | 3.47 | 4.52 |
| FlowSep (Orig., 1.7k h) | – | – | 1.78 | **0.70** | **0.72** | 0.14 | 2.99 | 4.76 | 3.03 | 4.38 |
| SAM-Audio (Orig., 1M h) | – | – | 4.27 | 1.05 | 0.40 | 0.21 | 3.14 | 3.75 | 3.72 | 3.60 |
| AudioSep (Hive, 2.4k h) | 7.61 | 6.43 | 0.69 | 0.64 | 0.71 | 0.22 | **3.44** | **4.91** | 3.48 | **4.54** |
| FlowSep (Hive, 2.4k h) | – | – | **1.76** | 0.81 | 0.69 | **0.18** | 3.18 | 4.85 | **3.79** | 4.47 |

CPU time per second of audio) are an order of magnitude or more costlier. Hive narrows the quality gap without changing this profile: AudioSep (Hive) recovers competitive perceptual scores against SAM-Audio at the AudioSep cost point, suggesting that purified supervision, rather than additional inference compute, is the operative lever for efficient query-based USS.

### 6.4. Performance Comparison on Third Party Test Sets

We further assessed out-of-distribution generalization on three independent benchmarks: USS-Bench, MUSDB18-HQ, and VGGClean_eval. Table 5 contrasts Hive-trained models with baselines trained on their original in-the-wild corpora. With only 2.4k hours of source audio, roughly one sixth of the 14.1k hours used to train the original AudioSep, AudioSep (Hive) raises USS-Bench SDR from $-1.86$ to 2.29 dB and OQ from 2.97 to 3.56, and improves MUSDB18-HQ SDR from $-1.01$ to 1.36 dB. On VGGClean_eval, both Hive-trained variants attain higher reference-free OQ (AudioSep: 3.44 vs. 3.42; FlowSep: 3.18 vs. 2.99), while the SDR of AudioSep (Hive) falls below AudioSep (Orig.) (7.61 vs. 8.67). This trade-off is consistent with the construction of VGGClean_eval, whose references can still contain co-occurring events, so suppressing such interference lowers reference-based SDR even as perceptual quality improves. The same pattern persists when we benchmark against SAM-Audio, a foundation model trained on approximately one million hours of audio: with less than 0.3% of that training budget, Hive-trained models remain competitive with, and in several cases exceed, SAM-Audio

*Table 6.* Mean shortcut gap on the controlled paired evaluation. Co-occurring and decorrelated mixtures share the same source count, target clip, and SNR vector, and differ only in interferer identities. Values are averaged over 2- to 5-source mixtures. The full PMI definition, decorrelated selection rule, paired construction protocol, and per-source-count breakdowns are provided in Appendix K.

*(a)* AudioSep (SDR ↑).

| Model | Decorr. | Co-occ. | Δ |
|---|---|---|---|
| AudioSep (Orig.) | 1.65 | 3.06 | $-1.41$ |
| AudioSep (Hive) | 5.48 | 5.87 | $-0.39$ |

*(b)* FlowSep (OQ ↑ and CLAP-T ↑).

| Model | OQ | | | CLAP-T | | |
|---|---|---|---|---|---|---|
| | Decorr. | Co-occ. | Δ | Decorr. | Co-occ. | Δ |
| FlowSep (Orig.) | 2.65 | 2.87 | $-0.23$ | 0.14 | 0.18 | $-0.04$ |
| FlowSep (Hive) | 3.00 | 3.09 | $-0.10$ | 0.18 | 0.20 | $-0.02$ |

under current USS benchmarks.

### 6.5. Controlled Shortcut Analysis

Aggregate gains on the Hive test set may conflate two effects, namely improved separation under semantically consistent mixtures and reduced reliance on co-occurrence shortcuts. To isolate the latter, we constructed a paired evaluation in which mixture complexity, target source, and SNR vector were held fixed, and only the statistical relation between the target and its interferers was manipulated. Co-occurring interferers are high-PMI AudioSet partners of the target, whereas decorrelated interferers have near-zero PMI under a minimum support constraint. Both groups satisfy the

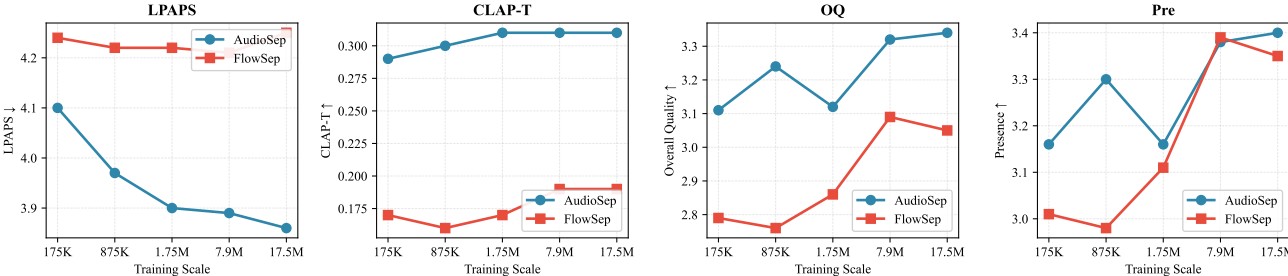

*Figure 6.* Scaling trends of AudioSep and FlowSep on the Hive test set across logarithmically increasing training data volumes (175k to 17.5M).

semantic compatibility matrix used by Hive, and acoustic-similarity outliers (target-to-interferer CLAP cosine above 0.8) were excluded so that the manipulation was restricted to statistical co-occurrence rather than acoustic resemblance. The full PMI definition, selection rules, and pair construction protocol are deferred to Appendix K.

Table 6 summarises the mean shortcut gap. AudioSep trained on its original AudioSet and VGGSound corpus exhibits a 1.41 dB SDR drop from co-occurring to decorrelated mixtures. After Hive training the gap shrinks to 0.39 dB, and the absolute SDR rises in both conditions, indicating that the improvement is not bought by sacrificing the easier setting. FlowSep displays the same pattern in perceptual and semantic metrics, with the OQ gap reducing from 0.23 to 0.10 and the CLAP-T gap from 0.04 to 0.02. Because each pair controls target identity, source count, and SNR, the residual gaps cannot be explained by changes in mixture difficulty or test distribution; they reflect the model's dependence on which interferer happens to be present. The narrowed gaps therefore indicate that Hive's purified single-event supervision reduces co-occurrence shortcut reliance rather than merely shifting the evaluation distribution. Per-source-count breakdowns in Table A5 and Table A6 show that this gap reduction is consistent across 2- to 5-source mixtures.

### 6.6. Impact of Training Data Scale

To investigate the marginal effect of training data scale on general audio separation performance and the scaling behavior of the Hive dataset, we constructed training subsets with logarithmic growth ranging from 175k to 17.5M samples and trained AudioSep under identical hyperparameter settings. As shown in Figure 6 and detailed in Table A7, increasing the training data consistently leads to marked and stable improvements across key metrics, including signal fidelity and perceptual quality. Specifically, when the training set size expands from 175k to the full 17.5M samples, the model achieves a 1.55 dB gain in SDR on the Hive test set. Meanwhile, the FAD metric, which reflects the naturalness of generated audio, improves from 0.85 to 0.80, and the semantic alignment metric CLAP-A score steadily rises to

0.65. This consistent performance improvement indicates that the Hive dataset exhibits high information density and diversity, supporting that continued model learning without saturation or overfitting at the current experimental scale. For a detailed analysis of the scaling behavior, please refer to Appendix L.

More importantly, these results strongly support the core argument that the purity of supervisory signals is more critical than data volume for general separation tasks. A cross comparison with baseline results in Table 4 shows that a model trained on only 875k samples (approximately 1,000 hours) already achieves an SDR of 4.96 dB, substantially surpassing the original AudioSep model trained on 14,100 hours of large-scale in-the-wild data from AudioSet and VGGSound (2.37 dB). This counterintuitive observation highlights that label-signal misalignment, which is prevalent in in-the-wild data, seriously limits model optimization. In contrast, the Hive dataset eliminates co-occurrence noise through strict semantic consistency constraints, enabling models to learn more precise source features and semantic mappings from significantly fewer training samples.

## 7. Conclusions

In this paper, we propose an automated data cleaning and synthesis pipeline and constructed Hive, a large-scale, high-fidelity synthetic dataset. By combining rigorous ontology reconstruction with semantic filtering powered by multi-modal large models, we successfully mined high-purity single-event segments from complex natural acoustic environments and synthesized training mixtures via a semantically consistent strategy. Extensive experiments demonstrate that models trained on the Hive dataset achieve separation accuracy and perceptual quality competitive with million-hour-scale SAM-Audio, while utilizing a data volume only ∼0.2% that of SAM-Audio. These findings validate that prioritizing purity of supervised signals offers a data-efficient alternative to brute-force scaling. By significantly reducing the data and computational resources required for training, this work provides a reproducible paradigm for developing robust USS.

## Acknowledgements

This work was supported in part by the National Key Research and Development Program of China (No. 2021ZD0200301), the National Natural Science Foundation of China (No. 62576187), and the Fundamental and Interdisciplinary Disciplines Breakthrough Plan of the Ministry of Education of China (No. JYB2025XDXM504).

## Impact Statement

This paper presents work whose goal is to advance the field of computational auditory scene analysis, and in particular query-based universal sound separation. Our contribution is methodological and infrastructural: a data-cleaning and synthesis pipeline together with the resulting Hive dataset, designed to enable robust open-domain separation models to be trained with substantially less data and compute than current million-hour-scale systems. By improving data efficiency, this line of work can lower the energy and infrastructure cost of training auditory foundation models, broaden access for groups without large-scale computing budgets, and support downstream applications such as immersive audio rendering, assistive listening, and on-device machine hearing.

We also recognize potential negative societal consequences. Improved universal sound separation can be misused to extract specific speakers or other sensitive acoustic cues from mixed recordings, and to support deceptive audio editing or repurposing of copyrighted content. In addition, our pipeline relies on multimodal large language models for relabeling and semantic compatibility estimation, which can propagate or amplify biases present in those models, and synthetic mixtures may not fully reflect the variability of real-world acoustic environments. Future work should incorporate room-impulse-response augmentation, naturally recorded benchmarks, explicit bias audits of the LLM-derived relabeling and co-occurrence matrices, and tail-class-aware sampling. Accordingly, we will release the dataset, pipeline code, and trained checkpoints under licenses that disallow surveillance and non-consensual identification, and we encourage downstream users to perform task-specific bias and misuse evaluations before deployment in high-stakes settings.

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

# A. Detailed Taxonomy Refinement for AudioSet Labels

In this section, we introduce the principles and process underlying our systematic reconstruction of the AudioSet label hierarchy (Gemmeke et al., 2017). While AudioSet covers a comprehensive range of concepts from specific sound sources to abstract acoustic attributes, this breadth introduces semantic ambiguity and discriminative conflicts in the query-based USS task. Based on the characteristics of source separation, we proposed a three-fold label refinement strategy to construct a semantically compact, hierarchically clear, and separable-source-oriented taxonomy. Furthermore, five professional audio practitioners were recruited to cross-validate this process.

**Merging and normalization of synonyms.** The AudioSet contains pairs of labels that are semantically equivalent but expressed differently, such as "Applause" and "Clapping", "Bow-wow" and "Bark", "Meow" and "Cat", "Moo" and "Cattle", and "Oink" and "Pig". The presence of these synonyms increases the dimensionality of the label space and causes implicit dispersion of training data. For example, "Applause" and "Clapping" correspond to the same acoustic event, yet the model must learn two essentially identical concepts. To eliminate this redundancy, we merged semantically equivalent label pairs and adopted the more representative or acoustically essential label as the standard form. For instance, we retained "Clapping" while merging "Applause", retained "Bark" while merging "Bow-wow", retained "Cat" while merging "Meow" and "Roar" (for felines), retained "Cattle" while merging "Moo", and retained "Boat" while merging "Ship". Notably, some synonym merged involve cross-level semantic consolidation; for example, "Ringtone" was merged into "Telephone bell ringing", "Train horn" and "Train whistle" were merged into "Train", and "Toot" was merged into "Vehicle horn/Honking".

**Hierarchical aggregation of fine-grained categories.** We observed numerous overly fine-grained leaf nodes in AudioSet with minimal differences in acoustic texture, such as "Accelerating, revving, vroom" vs. "Engine", "Acoustic guitar" vs. "Guitar", and "Alto saxophone" vs. "Saxophone". Although semantically distinct, these labels often exhibit highly overlapping distributions in the spectral feature space, making it difficult for classifiers to establish stable decision boundaries. Moreover, in source separation scenarios, users typically focus on the broad category of the source rather than subtle subtype differences; for instance, separating "guitar sound" is more practical than distinguishing between "acoustic guitar" and "electric guitar". Consequently, we unified these fine-grained labels into their parent nodes or semantic superordinates. For example, "Baby cry, infant cry" was merged into "Crying, sobbing"; "Bassoon" was merged into "Wind instrument, woodwind instrument"; and "Bugle", "Cornet", and "French horn" are consolidated into "Brass instrument". We adhered to the original AudioSet hierarchy during aggregation to maintain semantic integrity and logical consistency.

**Systematic exclusion of abstract acoustic attributes.** Unlike traditional audio event classification, the core objective of query-based USS is to extract sound source entities with specific physical origins from mixed signals rather than describing global attributes or environmental features. AudioSet contains many abstract labels describing physical or perceptual attributes, spatial or acoustic environments, technical media, or functional roles, such as "Inside, small room", "Outside, rural or natural", "Reverberation", "Echo", "Background music", "Soundtrack music", "Field recording", and "MP3". While valuable for scene understanding or retrieval, these labels cause conceptual confusion in source separation as they often correspond to background noise, reverberation effects, or recording conditions rather than independent foreground events. For instance, "Inside, small room" describes spatial attributes, "Reverberation" characterizes wave propagation, and "Background music" indicates a functional role rather than acoustic content. These traits provide no direct aid in determining which sources are separable. Therefore, we systematically excluded all such abstract attribute labels, ensuring the retained set includes only physically explicit, independently active, and separable source categories. Specifically, we removed spatial environment labels (e.g., "Inside, large room or hall"), acoustic attribute labels (e.g., "Bass (frequency range)", "Echo"), technical media labels (e.g., "MP3", "Television"), functional role labels (e.g., "Background music"), and perceptual attribute labels (e.g., "Dance music", "Tinnitus").

*Table A1.* Summary of non-retained AudioSet leaf nodes grouped by exclusion criteria. The arrow ($\rightarrow$) denotes fine-grained categories aggregated into their parent concepts; ~~red strikethrough~~ indicates abstract acoustic attributes excluded from the taxonomy; and the equivalence symbol ($\equiv$) represents synonym merging operations.

| Group / Labels |
| --- |
| **Synonym merge** |
| Applause $\equiv$ Clapping; Bow-wow $\equiv$ Bark; Caw $\equiv$ Crow; Clunk $\equiv$ Thunk; Coo $\equiv$ Pigeon, dove; Ding-dong $\equiv$ Doorbell; Harmony $\equiv$ Chord; Hoot $\equiv$ Owl; Honk $\equiv$ Goose; Meow $\equiv$ Cat; Moo $\equiv$ Cattle; Oink $\equiv$ Pig; Ringtone $\equiv$ Telephone bell ringing; Roar $\equiv$ Cat; Rock and roll $\equiv$ Rock music; Ship $\equiv$ Boat; Tick-tock $\equiv$ Clock; Tick $\equiv$ Clock; Thunk $\equiv$ Thump, thud; Toot $\equiv$ Vehicle horn/Honking; Train horn $\equiv$ Train; Train whistle $\equiv$ Train; Yell $\equiv$ Shout |
| **Fine-grained aggregation** |

Accelerating, revving, vroom → Engine; Acoustic guitar → Guitar; Afrobeat → Music of Africa; Alto saxophone → Saxophone; Angry music → Music mood; Artillery fire → Gunshot, gunfire; Babbling → Speech; Baby cry, infant cry → Crying, sobbing; Baby laughter → Laughter; Bass drum → Drum; Bass guitar → Guitar; Bassline → Musical concepts; Bassoon → Wind instrument, woodwind instrument; Battle cry → Shout; Bay → Dog; Bellow → Shout; Belly laugh → Laughter; Bird flight, flapping wings → Bird; Birthday music → Music role; Blare → Onomatopoeia; Bleat → Goat; Bluegrass → Country; Booing → Human group actions; Bugle → Brass instrument; Busy signal → Telephone; Cacophony → Noise; Cap gun → Gunshot, gunfire; Caterwaul → Cat; Cellphone buzz, vibrating alert → Telephone; Chainsaw → Light engine (high frequency); Cheering → Human group actions; Carnatic music → Music of Asia; Chipmunk → Rodents, rats, mice; Chink, clink → Glass; Children shouting → Human group actions; Chirp tone → Sine wave; Chorus effect → Effects unit; Chord → Musical concepts; Clavinet → Electric piano; Chuckle, chortle → Laughter; Clip-clop → Clicking; Cluck → Chicken, rooster; Compact disc → Sound reproduction; Christmas music → Music role; Clickety-clack → Clicking; Crash cymbal → Cymbal; Cornet → Brass instrument; Computer keyboard → Typing; Croak → Frog; Cumbia → Music of Latin America; Crowing, cock-a-doodle-doo → Chicken, rooster; Dental drill, dentist's drill → Drill; Dial tone → Telephone; Drone music → Ambient music; Drum roll → Snare drum; Drum beat → Beat; Dub → Reggae; Drone → Musical concepts; Drum machine → Drum kit; Dubstep → Electronic music; Distortion → Background noise; Electric guitar → Guitar; Duck call (hunting tool) → Miscellaneous sources; Electric toothbrush → Toothbrush; Electronic dance music → Electronic music; Electro → Electronic music; Electronica → Electronic music; Electronic organ → Organ; Electronic tuner → Sound equipment; Engine knocking → Engine; Engine starting → Engine; Exciting music → Music mood; Firecracker → Fireworks; French horn → Brass instrument; Funk carioca → Music of Latin America; Flamenco → Music of Latin America; Gasp → Breathing; Giggle → Laughter; Fusillade → Gunshot, gunfire; Funny music → Music mood; Glockenspiel → Mallet percussion; Gobble → Turkey; Gospel music → Christian music; Fizz → Onomatopoeia; Grunge → Rock music; Gramophone record → Sound reproduction; Guitar amplifier → Sound equipment; Hammond organ → Organ; Gull, seagull → Bird; Gush → Pour; Grind → Surface contact; Growling → Canidae, dogs, wolves; Heart murmur → Heart sounds, heartbeat; Heavy engine (low frequency) → Engine; Heavy metal → Rock music; Hi-hat → Cymbal; Harmonic → Sine wave; Happy music → Music mood; Headphones → Sound reproduction; House music → Electronic music; Howl (wind) → Wind; Idling → Engine; Infrasound → Other sourceless; Howl → Canidae, dogs, wolves; Jet engine → Aircraft engine; Kwaito → Music of Africa; Jingle (music) → Music role; Kuduro → Music of Latin America; Lullaby → Music role; Lawn mower → Light engine (high frequency); Machine gun → Gunshot, gunfire; Loudspeaker → Sound reproduction; Loop → Musical concepts; Mantra → Chant; Mains hum → Background noise; Maraca → Rattle (instrument); Medium engine (mid frequency) → Engine; Marimba, xylophone → Mallet percussion; Mellotron → Synthesizer; Motorboat, speedboat → Boat, Water vehicle; Melody → Musical concepts; Mouse → Rodents, rats, mice; Music of Bollywood → Music of Asia; Musical note → Musical concepts; Narration, monologue → Speech; Nicker → Horse; Neigh, whinny → Horse; Oldschool jungle → Drum and bass; Noise music → Electronic music; Musical ensemble → Musical instrument; Opera → Classical music; Pant → Breathing; Pizzicato → Violin, fiddle; Pink noise → Noise; Packing tape, duct tape → Domestic sounds, home sounds; Psychedelic rock → Rock music; Propeller, airscrew → Aircraft engine; Patter → Rodents, rats, mice; Progressive rock → Rock music; Pulse → Other sourceless; Quack → Duck; Purr → Cat; Punk rock → Rock music;
Puff → Onomatopoeia; Raindrop → Rain; Rhodes piano → Electric piano; Rimshot → Snare drum; Radio → Sound reproduction; Ringing (of resonator) → Other sourceless; Rowboat, canoe, kayak → Boat, Water vehicle; Rustling leaves → Wind; Sailboat, sailing ship → Boat, Water vehicle; Sad music → Music mood; Salsa music → Music of Latin America; Scary music → Music mood; Sampler → Synthesizer; Scratching (performance technique) → Musical instrument; Single-lens reflex camera → Camera; Sidetone → Background noise; Shatter → Glass; Snicker → Laughter; Snort → Breathing; Snoring → Breathing; Soca music → Music of Latin America; Snort (horse) → Horse; Sonic boom → Boom; Screech → Brief tone; Soprano saxophone → Saxophone; Squawk → Bird vocalization, bird call, bird song; Speech synthesizer → Speech; Steel guitar, slide guitar → Guitar; Steelpan → Mallet percussion; Static → Background noise; Tabla → Drum; Strum → Guitar; Tape hiss → Background noise; Techno → Electronic music; Tapping (guitar technique) → Guitar; Telephone dialing, DTMF → Telephone; Tender music → Music mood; Telephone bell ringing → Telephone; Throat clearing → Cough; Thunder → Thunderstorm; Throbbing → Noise; Theme music → Music role; Timpani → Drum; Trance music → Electronic music; Trickle, dribble → Pour; Trombone → Brass instrument; Trumpet → Brass instrument; Twang → Brief tone; Typewriter → Typing; Vibraphone → Mallet percussion; UK garage → Electronic music; Velcro, hook and loop fastener → Domestic sounds, home sounds; Video game music → Music role; Whimper → Crying, sobbing; Whimper (dog) → Dog; Wheeze → Breathing; Vibration → Noise; Whoop → Shout; Wind chime → Chime; Wolf-whistling → Whistling; Yak → Cattle, bovinae; Yawn → Human voice; Wedding music → Music role; Yip → Dog; Zing → Onomatopoeia

15

**Abstract acoustic attribute**

~~Background music~~; ~~Bass (frequency range)~~; ~~Bass (instrument role)~~; ~~Cat communication~~; ~~Children playing~~; ~~Conversation~~; ~~Crowd~~; ~~Creak~~; ~~Chatter~~; ~~Dance music~~; ~~Field recording~~; ~~Environmental noise~~; ~~Echo~~; ~~Inside, large room or hall~~; ~~Inside, public space~~; ~~Inside, small room~~; ~~MP3~~; ~~Outside, rural or natural~~; ~~Outside, urban or manmade~~; ~~Rain on surface~~; ~~Reverberation~~; ~~Sound effect~~; ~~Soundtrack music~~; ~~Song~~; ~~Television~~; ~~Traditional music~~; ~~Swing music~~; ~~Tinnitus, ringing in the ears~~; ~~Wind noise (microphone)~~; ~~White noise~~

# B. Prompts for Semantic-Acoustic Alignment

In this section, we provide the detailed prompts utilized by Qwen3-Omni[5] for the semantic-acoustic alignment framework described in Section 3.2. To enhance the robustness of the automated annotation and mitigate hallucination, we employed a majority voting strategy during inference. Specifically, we set the generation temperature to 1.0 to encourage diversity and performed $N = 10$ independent generation passes for each audio sample. The final label was determined by selecting the most frequent prediction among the ten outputs.

## B.1. Task I: Instance-Level Purification

To filter out samples containing background noise, transient interference, or multi-event mixtures, we instructed Qwen3-Omni to act as an audio quality auditor. The model is required to explicitly identify whether the audio clip consists of a single, distinct acoustic event. The prompt design is presented in Figure A1.

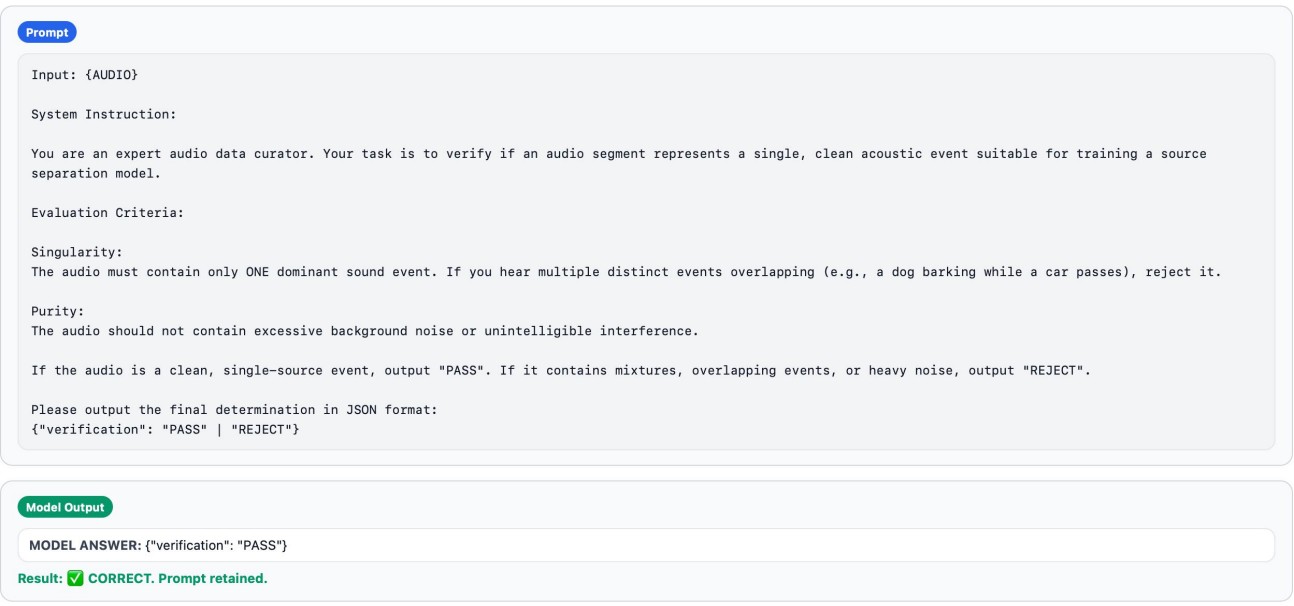

*Figure A1.* Prompt template used for Instance-level purification.

## B.2. Task II: Coarse-to-Fine Hierarchical Relabeling

For samples that passed the purification stage, we employed a coarse-to-fine strategy. Leveraging the coarse category predicted by the ZipFormer-based AudioTag model (e.g., "Domestic animals"), we constrained the search space for Qwen3-Omni to identify the specific leaf node in the ontology. This prompt injects the coarse prediction as a semantic prior. The prompt formulation is detailed in Figure A2.

---

[5]https://huggingface.co/Qwen/Qwen3-Omni-30B-A3B-Instruct

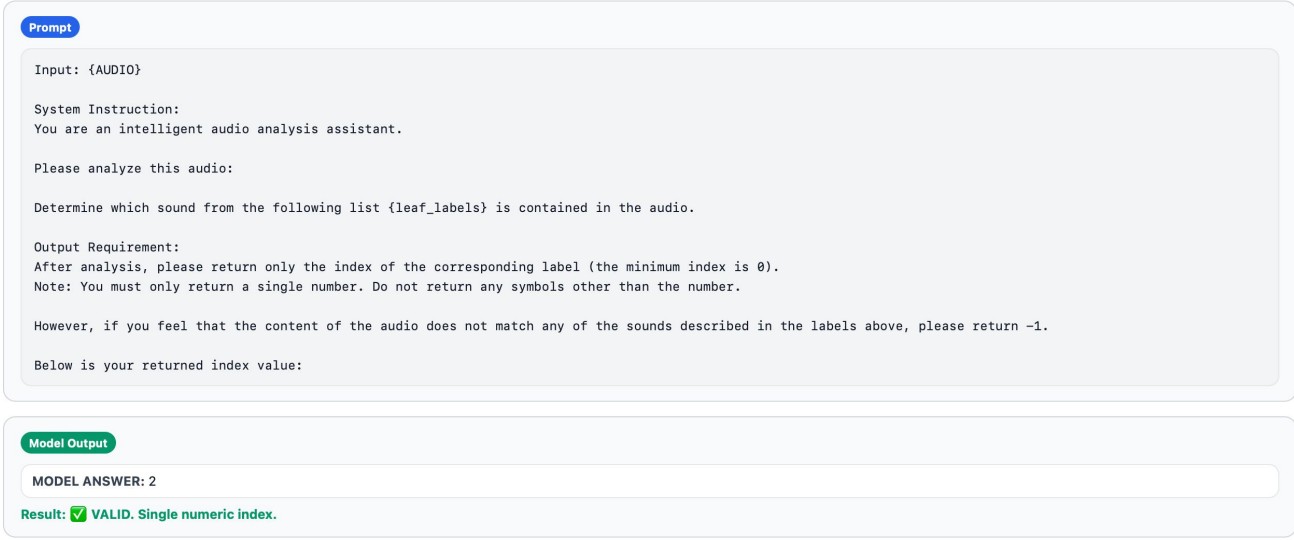

*Figure A2.* Prompt template used for hierarchical relabeling (leaf node selection). The variable {`leaf_labels`} is dynamically populated with the list of candidate leaf nodes derived from the coarse category (e.g., [`"Dog", "Cat", "Horse"`]). The model outputs an integer index or -1.

## B.3. Co-occurrence Validation Prompt

We provide the prompt template used to query Qwen3-Omni for judging whether two audio events can naturally co-occur in real-world recordings. Figure A3 shows the full prompt.

## B.4. Robustness Checks for LLM-based Annotation

To verify that the LLM-based stages are not overly sensitive to a particular prompt design, we performed additional robustness checks. For clip relabeling, Qwen3-Omni, Gemini 3.1 Pro, and GPT-Audio were compared under the same 4-AFC protocol described in Section 6.1. For semantic co-occurrence estimation, Qwen3.5 and GLM-4.7 achieved 94.7% and 94.9% agreement with Qwen3-Omni, respectively. Multiple prompt rephrasings yielded 98.4% overlap, indicating that the resulting decisions are stable across model and prompt variants.

# C. Source Datasets Details

*Table A2.* Summary of the twelve source datasets integrated into Hive. The collection spans diverse categories including general sound events, music, speech, and environmental soundscapes, all utilized under compliant academic or creative commons licenses.

| Dataset | Primary Type | License Type | # Clips | Duration (h) |
|---|---|---|---|---|
| BBC Sound Effects (British Broadcasting Corporation, 1991) | Sound Effects / Ambience | Remix License (NC Use) | 369,603 | 1,020.62 |
| AudioSet (Gemmeke et al., 2017) | General Audio Events | CC BY | 326,890 | 896.61 |
| VGGSound (Chen et al., 2020) | General / Real-world | CC BY 4.0 | 115,191 | 319.10 |
| MUSIC21 (Zhao et al., 2019) | Musical Instruments | YouTube Standard | 32,701 | 90.28 |
| FreeSound (Fonseca et al., 2017) | Diverse Crowdsourced | CC0 / CC BY / CC BY-NC | 17,451 | 46.90 |
| ClothoV2 (Drossos et al., 2020) | Soundscapes / Captioning | Non-Commercial Research | 14,759 | 38.19 |
| Voicebank-DEMAND (Botinhao et al., 2016) | Clean Speech | CC BY 4.0 | 12,376 | 9.94 |
| AVE (Tian et al., 2018) | Audio-Visual Events | CC BY-NC-SA | 3,054 | 6.91 |
| SoundBible | Short Sound Effects | CC BY 4.0 | 2,501 | 5.78 |
| DCASE (Mesaros et al., 2019) | Acoustic Scenes | Academic Use | 1,969 | 5.46 |
| ESC50 (Piczak, 2015) | Environmental Sounds | CC BY-NC 3.0 | 1,433 | 1.99 |
| FSD50K (Fonseca et al., 2021) | General Audio Events | Creative Commons | 636 | 0.80 |
| **Total Aggregated** | **Heterogeneous Sources** | – | **898,564** | **2,442.60** |

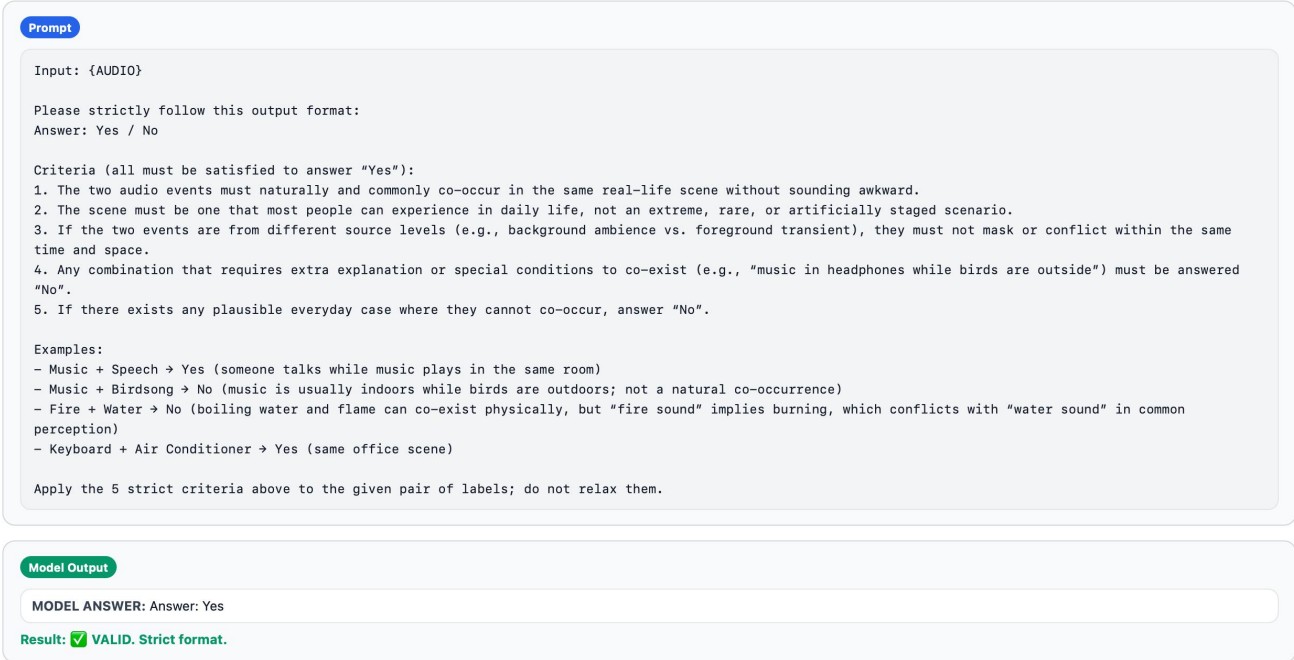

*Figure A3.* Prompt template used for evaluating the co-occurrence of audio events.

## D. Dataset Integration Details

To achieve broad coverage of real acoustic conditions and effectively capture rare events in the long-tailed distribution, we integrated high-quality audio resources from 12 public datasets, as shown in Table A2. All raw data were standardized and cleaned using our automated pipeline to remove silent segments and ensure label semantic consistency. The resulting pool contained $898,564$ independent audio files with a total duration of $2,442.60$ hours. Throughout the integration process, we strictly adhered to the license terms of each source dataset and utilized data solely within permitted academic or licensed scopes.

As the foundation of our collection, *BBC Sound Effects* (British Broadcasting Corporation, 1991) provides a professional sound effects library with extensive coverage. It mainly includes natural ambience, complex mechanical operations, and background sounds from daily environments, featuring broadcast-level recording fidelity. Under its noncommercial education and research license (Remix License), we selected 369,603 high-quality clips totaling 1,020.62 hours, which provide diverse acoustic event prototypes. *AudioSet* (Gemmeke et al., 2017)[6] is a widely used large-scale benchmark for audio event classification, largely composed of audio tracks from YouTube videos under the Creative Commons license (CC BY). Despite label noise in the raw data, its category coverage is essential for open-domain separation. After extensive cleaning, we utilized 326,890 clips with a total duration of 896.61 hours, forming another core pillar of our training data.

To increase the coverage of natural recordings, we included *VGGSound* (Chen et al., 2020)[7], which contains a large set of real-world audio sequences following the CC BY 4.0 license. We extracted 115,191 valid clips totaling 319.10 hours. For music, *MUSIC21* (Zhao et al., 2019)[8] provides key samples of solo and ensemble instruments. It follows YouTube copyright constraints and contains tracks that help the model learn complex harmonic structures. We incorporated 32,701 clips with a total duration of 90.28 hours to better distinguish music signals from broadband noise. We also leveraged crowdsourced audio from *FreeSound* (Fonseca et al., 2017)[9], uploaded by users worldwide and exhibiting substantial diversity in recording devices. After filtering files under CC0, CC BY, and CC BY-NC licenses, we retained 17,451 clips totaling 46.90 hours. This device heterogeneity was crucial for reducing overfitting to specific channel characteristics.

---

[6] https://huggingface.co/datasets/agkphysics/AudioSet
[7] https://huggingface.co/datasets/Loie/VGGSound
[8] https://github.com/roudimit/MUSIC_dataset
[9] https://huggingface.co/datasets/Meranti/CLAP_freesound

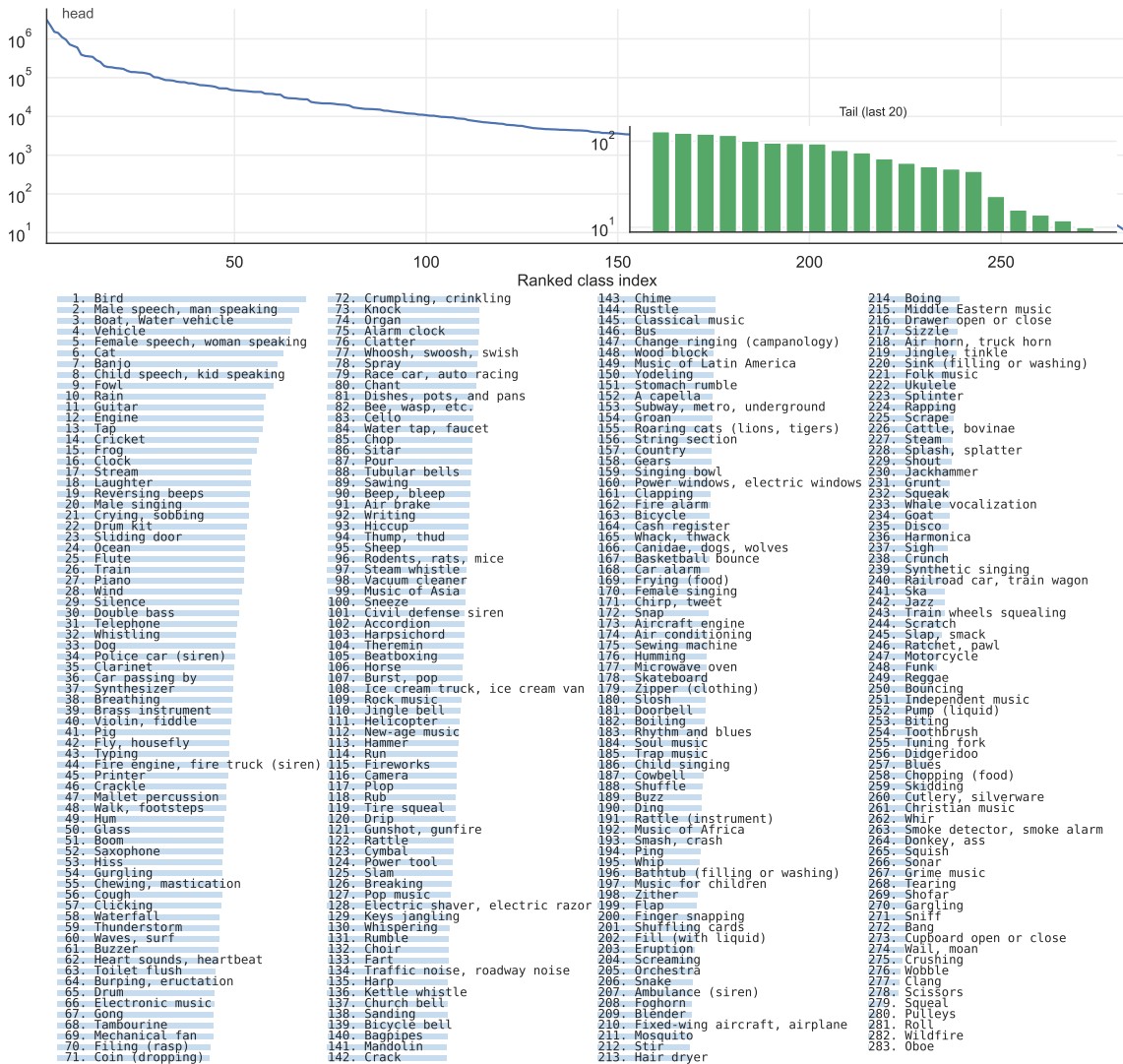

*Figure A4.* Complete frequency distribution and index mapping for the 283 labels in Hive. The top panel shows the rank frequency curve after sorting classes by decreasing frequency. The lower right inset zooms into the tail region (the last 20 classes) to highlight extremely low frequency categories. The bottom panel provides the full rank→label index table, where each class frequency is visually encoded by log normalized background bars.

For long-form audio description and complex soundscapes, *ClothoV2* (Drossos et al., 2020)[10] provides samples with rich temporal evolution. It is released under a noncommercial academic research license and is primarily used for audio captioning. After segmentation, we retained 14,759 clips totaling 38.19 hours. For speech processing, *Voicebank DEMAND* (Botinhao et al., 2016)[11] offers clean speech signals and is a standard dataset for speech enhancement. It follows CC BY 4.0 and contributed 12,376 clips totaling 9.94 hours, which were used to improve vocal separation clarity. For audio-visual event localization, *AVE (Audio Visual Event)* (Tian et al., 2018)[12] provides event samples with clear temporal boundaries. It is also based on YouTube and follows the CC BY-NC-SA license. After processing, the subset included 3,054 clips totaling 6.91 hours, supporting the modeling of transient event characteristics.

To supplement short sound effects for specific categories, we used *SoundBible* [13], which contains curated short audio clips and is primarily licensed under CC BY 4.0. Although smaller in scale, we acquired 2,501 clips totaling 5.78 hours; its high

---

[10]https://zenodo.org/records/4783391

[11]https://huggingface.co/datasets/JacobLinCool/VoiceBank-DEMAND-16k

[12]https://drive.google.com/open?id=1FjKwe79e0u96vdjIVwfRQ1V6SoDHe7kK

[13]https://huggingface.co/datasets/nyuuzyou/soundbible

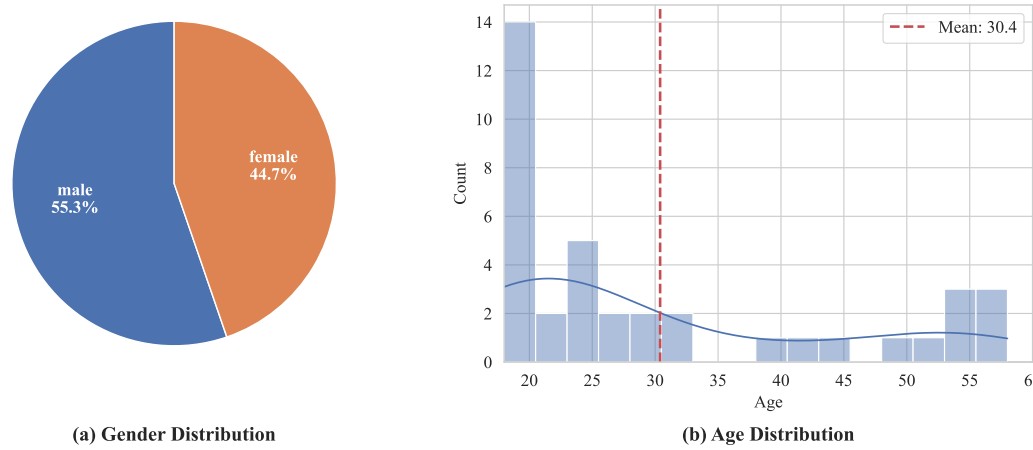

(a) Gender Distribution        (b) Age Distribution

*Figure A5.* Demographic distribution of participants. (a) Pie chart of gender ratio (male/female). (b) Histogram of age distribution, with the mean age indicated by a dashed line.

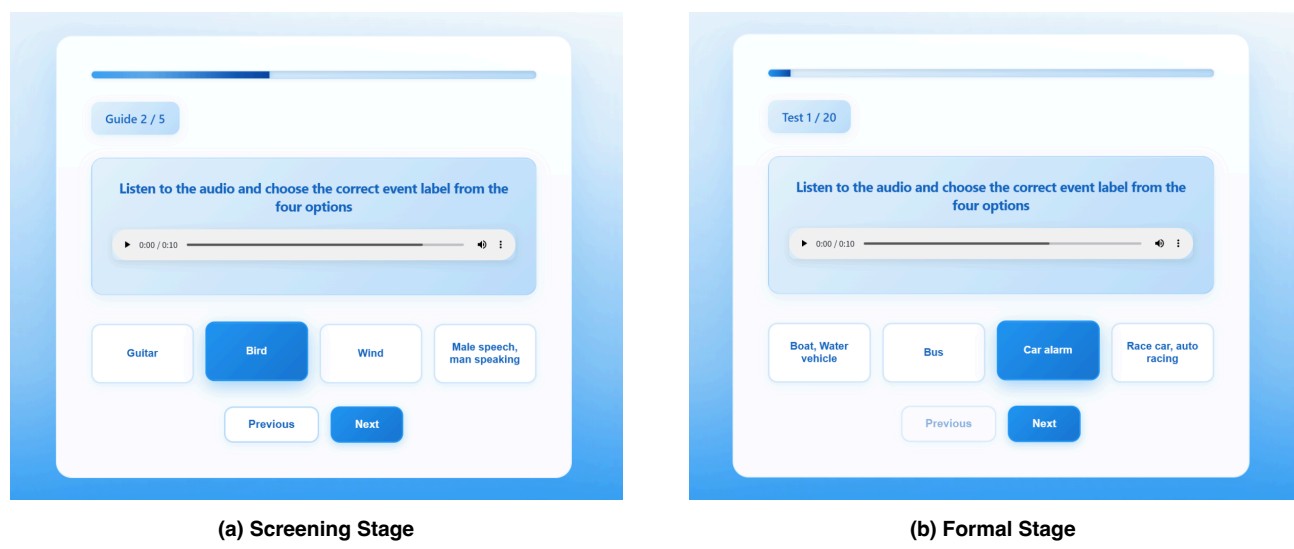

(a) Screening Stage        (b) Formal Stage

*Figure A6.* Annotation interface for the screening phase (A) and the main study phase (B).

signal-to-noise ratio offered clear supervision for rare events. For synthetic data, *DCASE* (Mesaros et al., 2019)[14], derived from acoustic scene detection challenges, provides synthetic and a small amount of real recordings. We employed 1,969 clips totaling 5.46 hours as complementary validation beyond real data. As a high-precision benchmark for environmental sound classification, *ESC50* (Piczak, 2015)[15] is a small manually curated dataset with highly accurate labels. It follows CC BY-NC 3.0 and yielded 1,433 clips totaling 1.99 hours, mainly for validating classification accuracy on standard environmental sound categories.

Finally, as a high-quality subset related to FreeSound, *FSD50K* (Fonseca et al., 2021)[16] provides finely annotated data based on the AudioSet ontology. We selected 636 additional clips totaling 0.80 hours to fill gaps in the long-tailed distribution. By integrating these twelve sources with compliant licensing and complementary characteristics, we constructed a long-tailed acoustic space that supports training for universal sound separation.

---

[14] https://zenodo.org/records/10886481
[15] https://huggingface.co/datasets/ashraq/esc50
[16] https://huggingface.co/datasets/Fhrozen/FSD50k

## E. Full Class Frequency Statistics of the Hive Dataset

In this section, we provide a detailed characterization of the scale and semantic distribution of the Hive dataset, complementing the main paper discussion on data diversity. The resulting Hive dataset serves as a large-scale benchmark with high semantic consistency. It comprises 19.6 million synthetic mixed audio samples, organized under a standard split into training, validation, and test sets, and covering highly concurrent mixtures from two-source to five-source settings. The dataset is built upon 283 fine-grained semantic classes, which ensures broad acoustic coverage while more faithfully reflecting the inherent long-tailed distribution observed in natural and man-made environments.

Our distribution analysis reveals pronounced class imbalance, which is visually evident in the full statistics shown in Figure A4. Among the 283 classes, head classes are dominated by frequently occurring natural sounds and human activities, such as *Bird* (approximately 3.115 million instances), *Male speech* (approximately 2.205 million instances), and transportation-related sounds such as *Boat* (approximately 1.523 million instances). These classes establish the primary supervision signal for model training. Simultaneously, the dataset preserves challenging long-tailed characteristics, with extremely sparse tail classes, including rare instrument sounds such as *Oboe* (only 10 instances), natural disaster sounds such as *Wildfire* (12 instances), and specific mechanical sound effects such as *Pulleys* (16 instances). This vast span from millions to tens of samples, covering six orders of magnitude, not only captures the complexity of open-world auditory scenes but also provides a strict and discriminative benchmark for evaluating the robustness of audio separation models under long-tailed distributions and their generalization to few-shot categories.

## F. Details of Out-of-Domain Datasets

To comprehensively evaluate the zero-shot generalization ability and robustness of AudioSep and FlowSep without fine-tuning, we employed three challenging out-of-domain test benchmarks: MUSDB18-HQ (Rafii et al., 2019), USS-Bench[17], and VGGClean_eval, the public VGGSound-based separation benchmark released by AudioSep (Liu et al., 2024). These benchmarks differ substantially from the Hive training set in acoustic characteristics and semantic distributions, enabling an effective assessment of practical performance on high-fidelity music separation, complex multi-source mixtures, and in-the-wild video audio.

MUSDB18-HQ is a widely used benchmark for music source separation. It contains 150 full-length songs across multiple genres, with a total duration of approximately 10 hours, and is split into 100 training songs and 50 test songs. All audio is stereo at 44.1 kHz. Unlike MUSDB18, which uses AAC compression with bandwidth limited to 16 kHz, MUSDB18-HQ provides uncompressed WAV files that preserve the full bandwidth up to 22 kHz, which is crucial for models that aim to recover high-frequency details. Each song includes a mixture and four isolated stems, namely drums, bass, vocals, and others. The tracks are collected from multiple high-quality sources, including DSD100, MedleyDB, the Native Instruments stems pack, and The Easton Ellises, and each mixture is the linear sum of its stems. For evaluation, we follow the standard protocol and use the `museval` toolkit to compute metrics such as the Signal-to-Distortion Ratio (SDR), which quantifies separation accuracy in complex music scenarios.

USS-Bench (Universal Sound Separation Benchmark) provides a more challenging wide-domain sound separation setting, primarily focusing on mixtures of speech and instruments. It combines Mandarin speech from AIShell-1 with instrument audio from the MUSIC dataset, forming mixture tasks with four sources where speech and instruments are randomly combined. USS-Bench includes both simulation data and real-world recordings, which are designed to test model adaptability under idealized and realistic acoustic conditions. The real-world recordings are collected in a single room using a microphone array with 2.8 cm spacing, where the distance between M1 and M4 is 8.4 cm, producing two-channel stereo signals and imposing stronger spatial resolving requirements on the model. We strictly follow the official validation set configuration with 297 valid samples and directly evaluate transfer from Hive without any additional fine-tuning, focusing on dense four-source mixtures and cross-microphone array signals.

VGGClean_eval is the public VGGSound-based separation benchmark released alongside AudioSep (Liu et al., 2024). It is constructed from the VGGSound test set (Chen et al., 2020), a large-scale collection of in-the-wild video clips with weak audio-visual labels. Following the original protocol, 100 clean samples were manually selected from the VGGSound test set, each containing a single distinct target sound event; this set of 100 samples is referred to as VGGSound-Clean. For each clean sample, 10 audio samples were randomly drawn from the remaining VGGSound test set to serve as interferers. Each pair of audio samples then has its loudness independently sampled uniformly between $-35$ and $-25$ dB LUFS, after which

---

[17] https://mab.to/GsSVrdZK0dfQ1/us3

the two signals are summed; if clipping occurs, the resulting mixture is rescaled so that the peak amplitude does not exceed 0.9. This procedure yields an evaluation set of 1,000 mixtures with an average SNR of approximately 0 dB. We adopted the released evaluation split without modification, used the provided text queries, and evaluated transfer from Hive without any additional fine-tuning. Because VGGSound clips can still contain residual co-occurring events that cannot be fully removed from the reference signal, we reported both reference-based metrics (e.g., SDR) and reference-free perceptual metrics on this benchmark, and treated the latter as the primary evidence of separation quality.

# G. Details of Baselines and Evaluation Metrics

## G.1. Baseline Methods

We evaluate a set of recent methods on the Hive test set. These baselines fall into two categories, discriminative models and generative models. The former typically separates sources by estimating masks or filters, while the latter synthesizes the target signal based on learned priors.

LASS-Net (Liu et al., 2022) is an early attempt for language queried universal audio source separation. It adopts a query-based separation framework consisting of a query network and a separation network. Specifically, it uses BERT to extract semantic embeddings from natural language descriptions and employs a ResUNet-based architecture to predict spectrogram masks conditioned on these embeddings. By jointly modeling acoustic and language information in an end-to-end manner, LASS-Net can separate a target source given arbitrary text queries, beyond a predefined label set.

CLIPSep (Dong et al., 2023) addresses data scarcity in text-queried separation by leveraging the shared embedding space of contrastive language image pretraining. Unlike supervised methods that require paired audio-text annotations, CLIPSep is trained on unlabeled noisy videos using visual queries extracted from video frames. At inference time, due to modality alignment in CLIP, it can transfer to text-based queries in a zero-shot manner. It uses a query vector modulation mechanism, where image or text embeddings modulate an audio separation network to bridge visual and auditory modalities for universal sound separation.

AudioSep (Liu et al., 2024) is a foundation model for open-domain sound separation. It expands the training paradigm by leveraging large-scale multimodal datasets and integrating a contrastive language-audio pretraining model as the text encoder. AudioSep adopts a frequency domain ResUNet separation backbone and predicts magnitude masks and phase residuals conditioned on CLAP text embeddings. Training on diverse datasets provides strong zero-shot generalization, enabling separation of a wide range of audio events, instruments, and speech described in natural language.

OmniSep (Cheng et al., 2025) proposes a unified all-modality sound separation framework that supports text, audio, and visual queries. Its key design is a Query Mixup training strategy that mixes query features from different modalities during training to encourage a unified representation space. It uses ImageBind to extract cross-modality aligned features and introduces a negative query mechanism to suppress undesired interference. This design enables joint optimization across modalities and robust separation performance regardless of the query modality at inference time.

ZETA (Manor & Michaeli, 2024) presents a zero-shot text-based method for audio editing and separation without task-specific training. It leverages generative priors from pretrained audio diffusion models such as AudioLDM. ZETA first applies DDPM inversion to map an input mixture into the latent noise space, and then performs guided denoising conditioned on the target text description. This approach repurposes generative diffusion models for discriminative separation, enabling extraction or manipulation of specific sources using text prompts.

FlowSep (Yuan et al., 2025) is a generative method based on rectified flow matching. Unlike diffusion models that simulate stochastic denoising trajectories, FlowSep constructs linear trajectories between a prior noise distribution and the data distribution in the latent space of a variational autoencoder. It uses a FLAN T5 encoder for text queries and conditions a flow matching network to generate latent features of the target source. By learning deterministic flow trajectories, FlowSep achieves high-quality separation and improves theoretical properties and inference efficiency compared with standard diffusion baselines.

ZeroSep (Huang et al., 2025) investigates training-free separation using pretrained text-guided audio diffusion models without fine-tuning. It relies on two generative inference steps, latent inversion and text conditional denoising. The mixture is first inverted into the latent space to capture composite acoustic information, and then denoising guided by a source-specific text prompt reconstructs the isolated signal. ZeroSep shows that strong generative priors can enable effective source separation in open-set scenarios, challenging the assumption that supervised training is necessary.

DGMO (Lee et al., 2025b), diffusion guided mask optimization, proposes a test-time optimization framework that repurposes pretrained diffusion models for zero-shot separation. Instead of directly generating outputs, DGMO optimizes a learnable spectrogram mask to ensure accurate temporal alignment with the input mixture, guided by the diffusion score function. This hybrid design mitigates phase inconsistency and hallucinated artifacts that are common in purely generative separation, improving signal fidelity.

SAM-Audio (Shi et al., 2025) is a foundation model for universal audio separation that integrates text, visual, and temporal span prompts. It is built on a diffusion Transformer architecture and trained via flow matching. SAM-Audio formulates separation as conditional generation and supports flexible prompting, including natural language, visual masks that segment target objects in videos, and time segments. The model operates in a compressed latent space and is trained on large-scale datasets covering speech, music, and environmental sounds. By effectively fusing multimodal cues, it achieves state-of-the-art source separation performance.

For training-on-Hive comparisons, we focus on AudioSep and FlowSep because they are reproducibly trainable representatives of the discriminative and generative paradigms. AudioSep is the strongest trainable discriminative baseline in signal fidelity among the evaluated methods, while FlowSep is the available generative baseline with a complete open training framework.

### G.2. Evaluation Metrics

Our evaluation protocol assesses separated audio from three complementary perspectives: signal fidelity, perceptual quality, and semantic alignment.

To measure signal fidelity, we used standard reference-based objective metrics, including Source-to-Distortion Ratio (SDR) (Vincent et al., 2006)[18] and Scale-Invariant Source-to-Distortion Ratio (SI-SDR) (Le Roux et al., 2019)[19]. These metrics require waveform-level alignment with the reference and therefore are most appropriate for discriminative, waveform-preserving separation models. For generative models, phase mismatch and fine-structure differences can make SDR and SI-SDR underestimate perceptual quality; we therefore report them only for discriminative models.

Because generative models may synthesize plausible high-frequency components that are not sample-aligned with the reference waveform, conventional signal metrics may not accurately reflect perceptual quality. We thus employ Fréchet Audio Distance (FAD) (Gui et al., 2024)[20] and Learned Perceptual Audio Patch Similarity (LPAPS) (Manor & Michaeli, 2024)[21]. FAD measures the distribution-level distance between generated audio and a reference set, capturing realism and the absence of artifacts, while LPAPS quantifies perceptual similarity in a deep feature space.

To verify semantic consistency between separated audio and the conditioning query, we use the CLAP score (Elizalde et al., 2023)[22]. We compute cosine similarity between the separated audio and the ground truth audio (CLAP Audio) to measure content preservation, and between the separated audio and the text description (CLAP Text) to measure query adherence. We further conducted a MUSHRA listening test on 50 representative samples covering the methods in Table 4, with 20 listeners and both a hidden reference and a low anchor, providing direct human evidence for perceptual quality and potential hallucination artifacts. Given the limitations of reference-based metrics for open-ended generation, we further adopt a large language model-based reference-free metric, SAM-Audio Judge (Shi et al., 2025)[23], which reports overall quality (OQ), recall (Rec), precision (Pre), and faithfulness (Fai). SAM-Audio Judge serves as secondary automated evidence rather than a dedicated hallucination-validation metric: for discriminative models, SDR and SI-SDR provide the primary reference-based check for hallucinated content, while for generative models, MUSHRA provides the main human validation.

---

[18] https://lightning.ai/docs/torchmetrics/stable/audio/signal_distortion_ratio.html
[19] https://lightning.ai/docs/torchmetrics/stable/audio/scale_invariant_signal_distortion_ratio.html
[20] https://github.com/HilaManor/AudioEditingCode/blob/codeclean/evals/fadtk_utils.py
[21] https://github.com/HilaManor/AudioEditingCode/blob/codeclean/evals/lpaps.py
[22] https://github.com/HilaManor/AudioEditingCode/blob/codeclean/evals/meta_clap_consistency.py
[23] https://huggingface.co/facebook/sam-audio-judge

# H. Details of 4-AFC Evaluation

## H.1. Evaluation Setup

We operationally defined semantic alignment as agreement between the selected label and the acoustic event present in the audio clip. Human consensus served as the reference label under a four-alternative forced-choice (4-AFC) protocol. Each trial contained a 10-second audio clip and four candidate event labels, and each evaluator, whether human or model, was required to select exactly one label. The evaluation set comprised 100 audio clips, yielding a random-guess accuracy of 25%. We further reported a class-frequency-stratified analysis to assess performance across category-frequency groups.

To reduce obvious cues and control task difficulty, each trial contained one target label and three distractor labels, where distractors were sampled from the same ontology level as the target. Concretely, for a clip with target label $y$, we sampled distractors $\{y_1^-, y_2^-, y_3^-\}$ from the label set that shared the same ontology depth as $y$, excluding $y$ itself. We also balanced candidate construction to mitigate frequency bias by monitoring distractor usage counts across the full evaluation set and prioritizing labels that were under-represented. The four candidates were then randomly ordered and presented in the same manner to all evaluators, including models. This yielded a shared retrieval-style multiple-choice setup for all humans, our pipeline, and all LALM baselines.

## H.2. Human Study

We recruited human participants online and retained 67 valid participants after screening. To improve the consistency of listening conditions, participants were required to wear headphones throughout the study and were allowed to replay each clip during annotation. An example annotation interface is shown in Figure A6. All participants provided informed consent and received compensation.

In the formal evaluation, each audio clip was independently annotated by all valid participants. We computed mean human accuracy by first obtaining each participant's accuracy across all clips and then averaging over participants. We further obtained a clip-level consensus label by majority voting over the annotations for the same clip. To quantify inter-rater agreement, we computed Fleiss' $\kappa$ over the 100 trials, treating the four candidate options as annotation categories. The resulting Fleiss' $\kappa$ of 0.843 indicated a high level of agreement.

## H.3. LALM Baseline Setup

We evaluated Qwen3-Omni, Gemini 3.1 Pro, and GPT-Audio under the same 4-AFC inputs presented to human participants, namely an audio clip together with four candidate labels. For comparability, every LALM evaluator was required to output exactly one option among the four, with no explanatory text. As shown in Figure A7, all models used an identical prompt template across all trials.

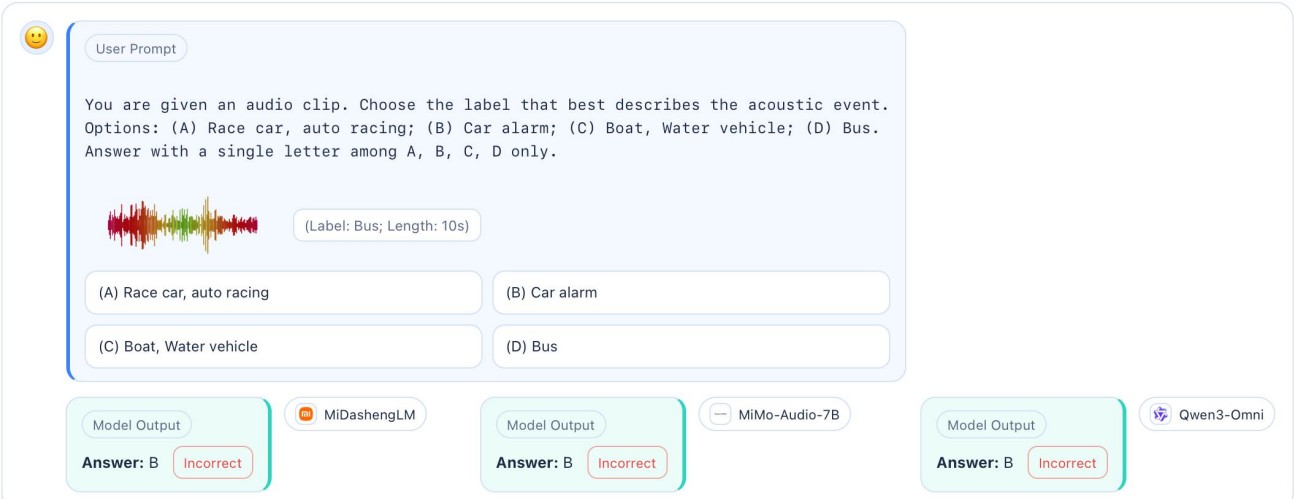

*Figure A7.* Prompt template and example trial for LALM evaluation. The model receives an audio clip and a multiple-choice question with four candidate labels, and must respond with a single letter (A/B/C/D).

*Table A3.* Detailed performance breakdown on the Hive test set across eight models and four mixture complexities.

| Model | Mix | Signal Fidelity ↑ | | Perceptual & Semantic Quality | | | | Reference-free Quality ↑ | | | |
|---|---|---|---|---|---|---|---|---|---|---|---|
| | | SDR | SI-SDR | LPAPS ↓ | FAD ↓ | CLAP-A ↑ | CLAP-T ↑ | OQ | Rec | Pre | Fai |
| *Discriminative Models* | | | | | | | | | | | |
| LASS-Net | 2-mix | 2.77 | 1.24 | 3.84 | 0.89 | 0.56 | 0.21 | 2.70 | 4.52 | 2.76 | 4.15 |
| | 3-mix | -2.26 | -3.28 | 4.67 | 1.02 | 0.45 | 0.16 | 2.48 | 4.47 | 2.53 | 4.11 |
| | 4-mix | -5.22 | -6.11 | 5.15 | 1.10 | 0.37 | 0.12 | 2.33 | 4.38 | 2.37 | 4.02 |
| | 5-mix | -7.15 | -8.01 | 5.45 | 1.15 | 0.32 | 0.08 | 2.24 | 4.33 | 2.28 | 3.96 |
| | **Overall** | **-2.97** | **-4.04** | **4.78** | **1.04** | **0.43** | **0.14** | **2.44** | **4.42** | **2.48** | **4.06** |
| CLIPSep | 2-mix | -2.08 | -6.23 | 5.52 | 0.95 | 0.52 | 0.17 | 2.83 | 4.27 | 3.11 | 3.49 |
| | 3-mix | -5.50 | -8.51 | 5.91 | 1.04 | 0.44 | 0.12 | 2.48 | 4.22 | 2.65 | 3.47 |
| | 4-mix | -7.85 | -10.64 | 6.13 | 1.09 | 0.39 | 0.08 | 2.29 | 4.06 | 2.44 | 3.32 |
| | 5-mix | -9.37 | -12.13 | 6.26 | 1.12 | 0.36 | 0.05 | 2.19 | 3.88 | 2.34 | 3.20 |
| | **Overall** | **-6.20** | **-9.38** | **5.95** | **1.05** | **0.43** | **0.10** | **2.45** | **4.11** | **2.63** | **3.37** |
| AudioSep | 2-mix | 9.55 | 8.31 | 3.14 | 0.70 | 0.72 | 0.32 | 3.44 | 4.87 | 3.46 | 4.52 |
| | 3-mix | 2.05 | 1.30 | 4.26 | 0.91 | 0.62 | 0.26 | 2.99 | 4.87 | 3.00 | 4.48 |
| | 4-mix | 0.80 | 0.18 | 4.42 | 0.95 | 0.53 | 0.26 | 2.74 | 4.85 | 2.75 | 4.45 |
| | 5-mix | -2.94 | -3.46 | 5.04 | 1.06 | 0.42 | 0.20 | 2.61 | 4.85 | 2.62 | 4.44 |
| | **Overall** | **2.37** | **1.58** | **4.22** | **0.90** | **0.57** | **0.26** | **2.94** | **4.86** | **2.96** | **4.47** |
| OmniSep | 2-mix | 1.57 | -0.91 | 4.27 | 0.82 | 0.63 | 0.21 | 2.91 | 4.63 | 3.05 | 3.88 |
| | 3-mix | -2.17 | -3.93 | 4.87 | 0.94 | 0.54 | 0.18 | 2.66 | 4.61 | 2.76 | 3.84 |
| | 4-mix | -4.61 | -6.17 | 5.19 | 1.01 | 0.47 | 0.14 | 2.54 | 4.56 | 2.64 | 3.80 |
| | 5-mix | -6.18 | -7.66 | 5.39 | 1.06 | 0.43 | 0.12 | 2.51 | 4.53 | 2.59 | 3.78 |
| | **Overall** | **-2.85** | **-4.67** | **4.93** | **0.96** | **0.52** | **0.16** | **2.65** | **4.58** | **2.76** | **3.82** |
| *Generative Models* | | | | | | | | | | | |
| ZETA | 2-mix | – | – | 5.37 | 1.03 | 0.45 | 0.26 | 2.74 | 4.02 | 3.59 | 3.00 |
| | 3-mix | – | – | 5.43 | 1.04 | 0.45 | 0.27 | 2.79 | 4.06 | 3.58 | 3.12 |
| | 4-mix | – | – | 5.45 | 1.04 | 0.44 | 0.28 | 2.85 | 4.13 | 3.52 | 3.26 |
| | 5-mix | – | – | 5.46 | 1.05 | 0.44 | 0.28 | 2.90 | 4.21 | 3.47 | 3.36 |
| | **Overall** | **–** | **–** | **5.43** | **1.04** | **0.44** | **0.28** | **2.82** | **4.10** | **3.54** | **3.19** |
| FlowSep | 2-mix | – | – | 3.50 | 0.72 | 0.68 | 0.19 | 2.93 | 3.98 | 3.07 | 3.66 |
| | 3-mix | – | – | 4.10 | 0.85 | 0.59 | 0.17 | 2.78 | 4.02 | 2.93 | 3.66 |
| | 4-mix | – | – | 4.45 | 0.93 | 0.53 | 0.15 | 2.69 | 4.08 | 2.83 | 3.70 |
| | 5-mix | – | – | 4.65 | 0.98 | 0.48 | 0.14 | 2.65 | 4.17 | 2.78 | 3.76 |
| | **Overall** | **–** | **–** | **4.18** | **0.87** | **0.57** | **0.16** | **2.76** | **4.06** | **2.90** | **3.69** |
| ZeroSep | 2-mix | – | – | 4.02 | 0.84 | 0.63 | 0.30 | 3.02 | 4.76 | 3.13 | 4.28 |
| | 3-mix | – | – | 4.46 | 0.91 | 0.57 | 0.26 | 2.92 | 4.77 | 3.02 | 4.28 |
| | 4-mix | – | – | 4.69 | 0.96 | 0.52 | 0.23 | 2.81 | 4.76 | 2.90 | 4.27 |
| | 5-mix | – | – | 4.84 | 1.00 | 0.49 | 0.21 | 2.72 | 4.75 | 2.81 | 4.26 |
| | **Overall** | **–** | **–** | **4.50** | **0.92** | **0.55** | **0.25** | **2.87** | **4.76** | **2.97** | **4.27** |
| DGMO | 2-mix | – | – | 4.52 | 0.92 | 0.56 | 0.29 | 2.88 | 4.50 | 3.26 | 3.75 |
| | 3-mix | – | – | 4.93 | 0.96 | 0.52 | 0.28 | 2.86 | 4.46 | 3.22 | 3.79 |
| | 4-mix | – | – | 5.10 | 0.99 | 0.49 | 0.27 | 2.84 | 4.48 | 3.18 | 3.87 |
| | 5-mix | – | – | 5.21 | 1.01 | 0.47 | 0.26 | 2.83 | 4.48 | 3.14 | 3.90 |
| | **Overall** | **–** | **–** | **4.94** | **0.97** | **0.51** | **0.27** | **2.85** | **4.48** | **3.19** | **3.85** |
| SAM-Audio | 2-mix | – | – | 4.60 | 0.91 | 0.52 | 0.20 | 3.16 | 3.81 | 3.63 | 3.69 |
| | 3-mix | – | – | 5.11 | 1.01 | 0.43 | 0.17 | 2.97 | 3.71 | 3.42 | 3.59 |
| | 4-mix | – | – | 5.45 | 1.08 | 0.37 | 0.14 | 2.79 | 3.74 | 3.22 | 3.60 |

Table A3 – continued from previous page

| Model | Mix | Signal Fidelity ↑ | | Perceptual & Semantic Quality | | | | Reference-free Quality ↑ | | | |
|---|---|---|---|---|---|---|---|---|---|---|---|
| | | SDR | SI-SDR | LPAPS ↓ | FAD ↓ | CLAP-A ↑ | CLAP-T ↑ | OQ | Rec | Pre | Fai |
| | 5-mix | – | – | 5.66 | 1.13 | 0.32 | 0.12 | 2.66 | 3.81 | 3.07 | 3.64 |
| | **Overall** | **–** | **–** | **5.21** | **1.03** | **0.41** | **0.16** | **2.90** | **3.77** | **3.33** | **3.63** |

## H.4. Analysis of Failure Cases

To better characterize model errors, we analyzed disagreements with human consensus in the 100-clip audit. Qwen3-Omni made two errors, both on ambiguous clips that were also missed by Gemini 3.1 Pro: Sample #20 (Train vs. Rain) and Sample #76 (Flute vs. Violin, fiddle). Human consensus was only 59.7% for both samples, indicating that the errors are concentrated in acoustically ambiguous rather than high-consensus clips.

The class-frequency analysis further supports this interpretation. Head classes such as Bird and Male speech, man speaking showed stronger consensus, while instrument and environmental categories such as Drum kit, Guitar, and Rain showed lower consensus or stronger secondary preferences. These cases motivate future improvements for tail and ambiguous categories, while suggesting no systematic degradation on high-consensus samples.

## I. Details of Zero-Shot Results

Table A3 provides a comprehensive breakdown of zero-shot separation performance across varying acoustic densities, ranging from standard two-source mixtures to highly complex five-source scenarios. This granular evaluation exposes specific failure modes in existing methods when facing acoustically dense environments driven solely by text labels.

A clear trend observed across all baselines is the significant performance degradation as the number of concurrent sources increases. For discriminative approaches, this is most evident in the sharp decline of SDR. For example, AudioSep achieves a high SDR in two-source mixtures but experiences a dramatic drop to negative values in five-source settings. This sign inversion suggests that in dense acoustic environments, the model struggles to effectively suppress interference, likely due to the reliance on co-occurrence biases present in its original training data which are absent in our semantically decorrelated test set. Similarly, early methods like CLIPSep fail to yield positive SDRs even in simple scenarios, indicating a fundamental inability to handle open-domain text queries without relying on contextual shortcuts found in uncurated data.

Regarding generative baselines such as SAM-Audio and FlowSep, we observe a distinct trade-off between perceptual plausibility and semantic fidelity. These models generally maintain lower FAD scores across all densities compared to discriminative counterparts, implying that they generate audio that sounds acoustically natural. However, their semantic adherence, measured by CLAP-Text similarity, deteriorates significantly in four-source and five-source scenarios. This discrepancy reveals that while generative models can synthesize high-fidelity audio textures, they are prone to semantic drift or hallucination in complex scenes, often generating plausible-sounding artifacts that do not align with the specific text query.

In contrast to these baselines, models trained on Hive demonstrate superior robustness in high-interference settings. As shown in Table 4, the Hive-trained AudioSep maintains positive signal fidelity even in the most challenging five-source scenarios. This resilience confirms that the proposed data construction pipeline effectively mitigates the co-occurrence noise and label-signal misalignment that typically severely limit models trained on larger but noisier in-the-wild datasets.

## J. Inference Efficiency Details

Table A4 reports parameter counts, MACs, CPU and GPU latency, and peak GPU memory for all baselines, measured on 1-second audio at 32 kHz and averaged over 1,000 runs after 5 warm-up iterations. The number of sampling steps for generative models follows the original paper settings. Latency is measured on a single NVIDIA A100 (80 GB) for GPU runs and a 32-core Intel Xeon for CPU runs; memory reflects peak allocator usage during a forward pass.

Discriminative models maintain a clear efficiency advantage. LASS-Net and AudioSep run within tens of milliseconds of GPU time and below 1.1 GB of GPU memory, making them practical for real-time or resource-constrained deployment. Conversely, generative methods incur substantial overhead. SAM-Audio's 8.2 B parameters require ∼32 GB of GPU

*Table A4.* Comparison of computational efficiency across audio separation models. CPU and GPU times are measured on 1-second audio, averaged over 1,000 runs after 5 warm-up iterations. The number of steps for generative models follows the original paper settings. Best results are **bold**, second-best are wavy underlined.

| Model | Params (M) ↓ | MACs (G/s) ↓ | CPU Time (s) ↓ | GPU Time (s) ↓ | GPU Mem (MB) ↓ |
|---|---|---|---|---|---|
| *Discriminative Models* | | | | | |
| LASS-Net (Liu et al., 2022) | **63.40** | **15.57** | **0.26** | **0.01** | **296.35** |
| CLIPSep (Dong et al., 2023) | 181.57 | 35.39 | 0.76 | 0.02 | 573.69 |
| AudioSep (Liu et al., 2024) | 238.60 | 120.94 | 1.99 | 0.02 | 1094.73 |
| OmniSep (Cheng et al., 2025) | 1231.09 | 48.97 | 1.12 | 0.08 | 4883.00 |
| *Generative Models* | | | | | |
| ZETA (300 steps) (Manor & Michaeli, 2024) | 420.98 | 27670.78 | 117.78 | 14.29 | 1730.68 |
| FlowSep (20 steps) (Yuan et al., 2025) | 693.59 | 20146.08 | 59.97 | 1.30 | 5851.35 |
| ZeroSep (100 steps) (Huang et al., 2025) | 420.98 | 9356.30 | 32.64 | 4.29 | 1723.50 |
| DGMO (300 steps) (Lee et al., 2025b) | 420.98 | 27714.31 | 4902.61 | 42.36 | 2642.30 |
| SAM-Audio (Shi et al., 2025) | 8248.14 | 720.43 | 65.30 | 1.32 | 32164.97 |

memory, exceeding consumer-grade hardware. Test-time optimization methods such as DGMO take $\sim 4.9 \times 10^3$ s of CPU time per second of audio, which precludes interactive use. Hive does not change this efficiency profile: AudioSep (Hive) inherits the original AudioSep cost while substantially closing the perceptual-quality gap to SAM-Audio (Section 6.4). Taken together with the data-volume comparison in Section 6.6, this indicates that the bottleneck for efficient query-based USS lies in supervision purity, not inference compute.

# K. Controlled Shortcut Experiment Details

This section details the construction of the paired evaluation summarised in Section 6.5. The goal is to disentangle two effects that are otherwise confounded on the standard Hive test set, namely improvements that arise simply because Hive contains semantically consistent mixtures, and improvements that arise because Hive training reduces reliance on co-occurrence shortcuts. To this end we hold mixture complexity, target identity and SNR vector fixed, and vary only the statistical relation between the target and its interferers.

**Pointwise Mutual Information.**  We quantify the statistical co-occurrence between two events $i$ and $j$ in AudioSet using Pointwise Mutual Information,

$$\mathrm{PMI}(i,j) = \log_2 \frac{P(i,j)}{P(i)\,P(j)}, \tag{1}$$

where $P(i,j)$ is the empirical fraction of AudioSet training clips whose multi-label annotation contains both $i$ and $j$, and $P(i)$, $P(j)$ are the corresponding marginal frequencies. Frequencies are estimated after mapping AudioSet labels onto Hive's 283-class ontology described in Section 4. A large positive PMI indicates that two events co-occur far more often than independent labelling would predict, which is the regime in which a separation model can rely on the presence of one event as a cue for the other. A PMI close to zero indicates that the two events occur essentially independently, while a negative PMI indicates an avoidance pattern. We use PMI rather than raw co-occurrence counts because raw counts are dominated by globally frequent labels such as Speech, which appear together with almost any other event simply by virtue of their high marginal frequency. Normalising by $P(i)P(j)$ removes this baseline effect and yields a measure of whether the two events are statistically tied.

**Selection of co-occurring and decorrelated interferers.**  For each target event we form two pools of admissible interferers. The co-occurring pool consists of the events with the highest PMI to the target, restricted to the subset for which the semantic compatibility matrix $\mathbf{M}$ used by Hive is one. The decorrelated pool consists of events whose PMI to the target is close to zero, restricted to the same compatibility constraint and additionally requiring a minimum support of ten co-occurring AudioSet clips. The minimum-support constraint is important because PMI values estimated from very few co-occurrences are statistically unstable and can attain near-zero values purely by chance; including such events would conflate genuine statistical independence with sampling noise from the long tail. Both pools therefore satisfy the same physical-plausibility constraint enforced during Hive construction, and differ only in the statistical relation between the target and its interferers.

*Table A5.* Per-source-count shortcut analysis for AudioSep. Each row pair shares target identity, source count and SNR vector across the co-occurring and decorrelated conditions. Lower $|\Delta|$ indicates weaker reliance on co-occurrence shortcuts.

| Model | Mix | Decorr. SDR↑ | Co-occ. SDR↑ | $\Delta$ |
|---|---|---|---|---|
| AudioSep (Orig.) | 2-mix | 9.02 | 10.14 | −1.12 |
| AudioSep (Orig.) | 3-mix | 1.33 | 2.69 | −1.36 |
| AudioSep (Orig.) | 4-mix | 0.06 | 1.48 | −1.42 |
| AudioSep (Orig.) | 5-mix | −3.80 | −2.08 | −1.72 |
| AudioSep (Hive) | 2-mix | 10.98 | 11.24 | −0.26 |
| AudioSep (Hive) | 3-mix | 6.34 | 6.70 | −0.36 |
| AudioSep (Hive) | 4-mix | 3.39 | 3.83 | −0.44 |
| AudioSep (Hive) | 5-mix | 1.22 | 1.70 | −0.48 |

**Paired construction protocol.** For each source count $k \in \{2, 3, 4, 5\}$ we construct one thousand paired mixtures. We first sample a target event $T$ from a list of approximately thirty Hive classes that span head and middle frequencies, including events such as Bird, Male speech, Rain, Dog, Guitar, Piano, Drum kit and Thunder. We require that each candidate target admit at least four eligible interferers in both the co-occurring and the decorrelated pool, so that the same target can populate every source count up to $k = 5$. A clean target clip is then drawn from Hive's purified single-event sources, and an SNR vector is sampled with each component drawn uniformly from $[-5, 5]$ dB, matching Hive's mixing protocol. The co-occurring mixture is formed by sampling $k - 1$ interferers from the co-occurring pool, drawing one purified clip per interferer, and overlaying all sources after duration normalisation to ten seconds and energy unification to $\mathrm{RMS} = 0.1$, with relative gains determined by the SNR vector. The decorrelated mixture reuses the same target clip and the same SNR vector, but draws its $k - 1$ interferers from the decorrelated pool instead. By construction the two mixtures in a pair share target identity, source count, energy normalisation and SNR allocation, and differ only in interferer identity. The final evaluation contains eight thousand mixtures in total, four thousand co-occurring and four thousand decorrelated.

**Acoustic similarity control.** A natural concern is that high-PMI partners of a target might also be acoustically more similar to it than near-zero-PMI partners, in which case a residual gap could be attributed to acoustic difficulty rather than to shortcut reliance. We monitor this confound using the CLAP audio-embedding cosine similarity between the target clip and each interferer clip. We exclude any mixture in which the maximum target-to-interferer similarity exceeds $0.8$, removing the most acoustically degenerate cases at the source. We further track the mean target-to-interferer similarity in each group as a covariate, and find the two groups to be closely matched after the threshold filter, so the observed gap between co-occurring and decorrelated conditions cannot be explained by a systematic acoustic-similarity offset. The remaining gap therefore reflects the model's behaviour as a function of statistical co-occurrence, with mixture complexity and acoustic similarity controlled.

**Per-source-count results.** Table A5 reports the AudioSep results in signal-fidelity SDR for each source count, and Table A6 reports the FlowSep results in perceptual quality OQ and semantic alignment CLAP-T. The mean shortcut gaps reported in the main text correspond to averaging the $\Delta$ columns over $k \in \{2, 3, 4, 5\}$. Across both models and all source counts, training on Hive consistently shrinks the gap between co-occurring and decorrelated conditions, with the absolute reduction growing as the mixture becomes more crowded. This trend is consistent with the interpretation that a model relying on statistical co-occurrence is most exposed in dense mixtures, where alternative cues are scarcest, and that purified single-event supervision specifically attenuates this dependence.

**Scope and limitations.** The thirty target classes used here cover head and middle frequencies of the Hive ontology, but exclude extremely rare tail events for which neither pool can be reliably populated. The PMI estimates inherit any bias present in the AudioSet labelling protocol, in particular the well-known under-labelling of background sounds. Finally, the analysis isolates statistical co-occurrence as a shortcut signal but does not preclude other shortcut sources, such as room acoustics or recording-channel artefacts, which would require an in-the-wild evaluation to characterise. We view the present experiment as direct causal evidence on one specific and well-defined shortcut, and as complementary to the aggregate Hive test-set results in Section 6.4.

*Table A6.* Per-source-count shortcut analysis for FlowSep. Each row pair shares target identity, source count and SNR vector across the co-occurring and decorrelated conditions. OQ denotes overall perceptual quality and CLAP-T denotes CLAP text-audio similarity. Lower $|\Delta|$ indicates weaker reliance on co-occurrence shortcuts.

| Model | Mix | Decorr. OQ↑ | Co-occ. OQ↑ | ΔOQ | Decorr. CLAP-T↑ | Co-occ. CLAP-T↑ | ΔCLAP-T |
|---|---|---|---|---|---|---|---|
| FlowSep (Orig.) | 2-mix | 2.84 | 3.02 | −0.18 | 0.17 | 0.21 | −0.04 |
| FlowSep (Orig.) | 3-mix | 2.67 | 2.88 | −0.21 | 0.15 | 0.19 | −0.04 |
| FlowSep (Orig.) | 4-mix | 2.57 | 2.81 | −0.24 | 0.13 | 0.17 | −0.04 |
| FlowSep (Orig.) | 5-mix | 2.51 | 2.78 | −0.27 | 0.11 | 0.16 | −0.05 |
| FlowSep (Hive) | 2-mix | 3.24 | 3.30 | −0.06 | 0.22 | 0.24 | −0.02 |
| FlowSep (Hive) | 3-mix | 3.00 | 3.09 | −0.09 | 0.19 | 0.21 | −0.02 |
| FlowSep (Hive) | 4-mix | 2.90 | 3.00 | −0.10 | 0.16 | 0.18 | −0.02 |
| FlowSep (Hive) | 5-mix | 2.85 | 2.98 | −0.13 | 0.14 | 0.16 | −0.02 |

## L. Scaling Law Analysis

In this section, we provide a granular analysis of how model performance evolves as the volume of high-purity training data increases logarithmically from 175k to 17.5M samples. By comparing a representative discriminative model (AudioSep) and a generative model (FlowSep), we observe distinct scaling laws that shed light on the different data requirements for signal estimation versus conditional synthesis.

We first examine the scaling dynamics of discriminative models. As detailed in Table A7, AudioSep exhibits a continuous, log-linear improvement in signal fidelity metrics without saturation. The SDR increases monotonically from 4.12 dB at 175k samples to 5.67 dB at 17.5M, suggesting that the model continuously benefits from the high informational density of additional clean samples to refine its separation masks. This advantage is most pronounced in complex scenarios; in the challenging 5-mix setting, the model transitions from near-failure (−0.05 dB SDR) at the smallest scale to robust separation (1.46 dB SDR) at the largest. Remarkably, even at an intermediate scale of 875k samples, AudioSep achieves an Overall SDR of 4.96 dB, surpassing the official baseline trained on millions of in-the-wild samples. This empirically validates that the purity of supervised signals is a more decisive factor than raw data volume for discriminative alignment.

In contrast, the generative FlowSep model reveals a more complex, two-phase scaling behavior, as shown in Table A8. Distance-based perceptual metrics such as LPAPS and FAD improve rapidly in the low-data regime but saturate early, with FAD stabilizing around 0.85 by the 1.75M scale. This indicates that the model learns to synthesize acoustically plausible textures with relatively limited data. However, semantic adherence and overall quality exhibit a critical mass effect. Reference-free metrics like OQ remain relatively flat between 175k and 1.75M but experience a sudden surge at the 7.9M scale, rising to 3.09. Similarly, the precision score increases significantly from 3.11 to 3.39. This implies that while the model quickly learns to mimic acoustic textures, it requires significantly larger-scale data to cross the threshold for accurate semantic control and to mitigate hallucinations in complex scenes.

This comparative analysis demonstrates that Hive supports efficient training for both paradigms through distinct mechanisms. For discriminative models, the dataset provides a steep, continuous gradient for optimizing signal fidelity. For generative models, the scale of Hive is sufficient to cross the usability threshold, enabling models to advance beyond mere acoustic mimicry to achieve high-fidelity, semantically consistent generation.

*Table A7.* Performance of AudioSep trained on Hive dataset with different training scales (175K, 875K, 1.75M, 7.9M, and 17.5M samples) across four mixture complexities on the Hive test set.

| Scale | Mix | Signal Fidelity ↑ | | Perceptual & Semantic Quality | | | | Reference-free Quality ↑ | | | |
|---|---|---|---|---|---|---|---|---|---|---|---|
| | | SDR | SI-SDR | LPAPS ↓ | FAD ↓ | CLAP-A ↑ | CLAP-T ↑ | OQ | Rec | Pre | Fai |
| 175K | 2-mix | 9.52 | 8.51 | 3.15 | 0.70 | 0.73 | 0.32 | 3.13 | 4.88 | 3.14 | 4.53 |
| | 3-mix | 4.94 | 4.19 | 4.00 | 0.83 | 0.63 | 0.30 | 3.13 | 4.84 | 3.17 | 4.43 |
| | 4-mix | 2.07 | 1.42 | 4.47 | 0.91 | 0.56 | 0.28 | 3.12 | 4.81 | 3.17 | 4.36 |
| | 5-mix | -0.05 | -0.65 | 4.79 | 0.97 | 0.51 | 0.26 | 3.09 | 4.83 | 3.15 | 4.36 |
| | **Overall** | **4.12** | **3.37** | **4.10** | **0.85** | **0.61** | **0.29** | **3.11** | **4.84** | **3.16** | **4.40** |
| 875K | 2-mix | 10.41 | 9.50 | 3.00 | 0.67 | 0.75 | 0.32 | 3.26 | 4.87 | 3.28 | 4.52 |
| | 3-mix | 5.83 | 5.13 | 3.86 | 0.80 | 0.65 | 0.31 | 3.27 | 4.83 | 3.31 | 4.42 |
| | 4-mix | 2.88 | 2.27 | 4.35 | 0.88 | 0.59 | 0.29 | 3.24 | 4.79 | 3.31 | 4.34 |
| | 5-mix | 0.72 | 0.15 | 4.68 | 0.94 | 0.54 | 0.27 | 3.21 | 4.80 | 3.28 | 4.32 |
| | **Overall** | **4.96** | **4.26** | **3.97** | **0.82** | **0.63** | **0.30** | **3.24** | **4.82** | **3.30** | **4.40** |
| 1.75M | 2-mix | 11.00 | 10.15 | 2.93 | 0.65 | 0.76 | 0.33 | 3.13 | 4.88 | 3.14 | 4.53 |
| | 3-mix | 6.45 | 5.74 | 3.78 | 0.79 | 0.67 | 0.32 | 3.13 | 4.84 | 3.17 | 4.43 |
| | 4-mix | 3.56 | 2.91 | 4.28 | 0.86 | 0.60 | 0.30 | 3.12 | 4.81 | 3.17 | 4.36 |
| | 5-mix | 1.41 | 0.79 | 4.61 | 0.92 | 0.55 | 0.28 | 3.09 | 4.83 | 3.15 | 4.36 |
| | **Overall** | **5.60** | **4.90** | **3.90** | **0.81** | **0.64** | **0.31** | **3.12** | **4.84** | **3.16** | **4.42** |
| 7.9M | 2-mix | 11.01 | 10.17 | 2.93 | 0.66 | 0.76 | 0.33 | 3.34 | 4.86 | 3.36 | 4.51 |
| | 3-mix | 6.50 | 5.82 | 3.78 | 0.79 | 0.66 | 0.32 | 3.34 | 4.81 | 3.40 | 4.40 |
| | 4-mix | 3.60 | 2.97 | 4.27 | 0.86 | 0.60 | 0.30 | 3.32 | 4.77 | 3.40 | 4.32 |
| | 5-mix | 1.42 | 0.83 | 4.60 | 0.92 | 0.55 | 0.28 | 3.29 | 4.78 | 3.38 | 4.29 |
| | **Overall** | **5.63** | **4.95** | **3.89** | **0.81** | **0.64** | **0.31** | **3.32** | **4.80** | **3.38** | **4.38** |
| 17.5M | 2-mix | 11.11 | 10.31 | 2.88 | 0.64 | 0.77 | 0.33 | 3.35 | 4.86 | 3.37 | 4.52 |
| | 3-mix | 6.52 | 5.86 | 3.75 | 0.78 | 0.67 | 0.32 | 3.36 | 4.81 | 3.41 | 4.41 |
| | 4-mix | 3.61 | 3.01 | 4.24 | 0.86 | 0.61 | 0.30 | 3.33 | 4.77 | 3.42 | 4.33 |
| | 5-mix | 1.46 | 0.90 | 4.57 | 0.91 | 0.56 | 0.28 | 3.31 | 4.78 | 3.40 | 4.30 |
| | **Overall** | **5.67** | **5.02** | **3.86** | **0.80** | **0.65** | **0.31** | **3.34** | **4.81** | **3.40** | **4.39** |

*Table A8.* Performance of FlowSep trained on Hive dataset with different training scales (175K, 875K, 1.75M, 7.9M, and 17.5M samples) across four mixture complexities on the Hive test set.

| Scale | Mix | Perceptual & Semantic Quality | | | | Reference-free Quality ↑ | | | |
|---|---|---|---|---|---|---|---|---|---|
| | | LPAPS ↓ | FAD ↓ | CLAP-A ↑ | CLAP-T ↑ | OQ | Rec | Pre | Fai |
| 175K | 2-mix | 3.47 | 0.69 | 0.72 | 0.21 | 2.97 | 4.39 | 3.13 | 3.86 |
| | 3-mix | 4.20 | 0.84 | 0.61 | 0.18 | 2.79 | 4.19 | 3.02 | 3.63 |
| | 4-mix | 4.55 | 0.92 | 0.55 | 0.15 | 2.70 | 4.11 | 2.95 | 3.56 |
| | 5-mix | 4.75 | 0.97 | 0.50 | 0.13 | 2.68 | 4.13 | 2.93 | 3.56 |
| | **Overall** | **4.24** | **0.86** | **0.60** | **0.17** | **2.79** | **4.21** | **3.01** | **3.65** |
| 875K | 2-mix | 3.41 | 0.68 | 0.73 | 0.21 | 2.94 | 4.39 | 3.09 | 3.91 |
| | 3-mix | 4.19 | 0.85 | 0.61 | 0.17 | 2.76 | 4.15 | 2.99 | 3.65 |
| | 4-mix | 4.54 | 0.93 | 0.54 | 0.14 | 2.68 | 4.04 | 2.94 | 3.55 |
| | 5-mix | 4.75 | 0.98 | 0.49 | 0.12 | 2.66 | 4.04 | 2.92 | 3.54 |
| | **Overall** | **4.22** | **0.86** | **0.59** | **0.16** | **2.76** | **4.16** | **2.98** | **3.66** |
| 1.75M | 2-mix | 3.43 | 0.68 | 0.73 | 0.22 | 3.06 | 4.40 | 3.23 | 3.94 |
| | 3-mix | 4.18 | 0.84 | 0.62 | 0.18 | 2.87 | 4.16 | 3.12 | 3.68 |
| | 4-mix | 4.53 | 0.92 | 0.55 | 0.15 | 2.77 | 4.05 | 3.06 | 3.57 |
| | 5-mix | 4.74 | 0.97 | 0.50 | 0.13 | 2.75 | 4.04 | 3.04 | 3.56 |
| | **Overall** | **4.22** | **0.85** | **0.60** | **0.17** | **2.86** | **4.16** | **3.11** | **3.69** |
| 7.9M | 2-mix | 3.46 | 0.67 | 0.73 | 0.24 | 3.33 | 4.43 | 3.54 | 4.00 |
| | 3-mix | 4.15 | 0.81 | 0.63 | 0.20 | 3.11 | 4.18 | 3.41 | 3.73 |
| | 4-mix | 4.51 | 0.89 | 0.57 | 0.17 | 2.98 | 4.06 | 3.33 | 3.61 |
| | 5-mix | 4.72 | 0.94 | 0.53 | 0.15 | 2.93 | 4.05 | 3.29 | 3.60 |
| | **Overall** | **4.21** | **0.83** | **0.62** | **0.19** | **3.09** | **4.18** | **3.39** | **3.73** |
| 17.5M | 2-mix | 3.52 | 0.69 | 0.72 | 0.23 | 3.27 | 4.43 | 3.48 | 3.96 |
| | 3-mix | 4.21 | 0.82 | 0.63 | 0.20 | 3.05 | 4.18 | 3.35 | 3.69 |
| | 4-mix | 4.54 | 0.90 | 0.57 | 0.17 | 2.95 | 4.06 | 3.29 | 3.58 |
| | 5-mix | 4.74 | 0.95 | 0.53 | 0.15 | 2.92 | 4.04 | 3.27 | 3.58 |
| | **Overall** | **4.25** | **0.84** | **0.61** | **0.19** | **3.05** | **4.18** | **3.35** | **3.70** |

