# OpenReview forum: "A Semantically Consistent Dataset for Data-Efficient Query-Based Universal Sound Separation"
_ICML.cc/2026/Conference — ICML 2026 regular_

### Official Review · Reviewer_gXXQ · 2026-03-11

**Soundness:** 4
**Presentation:** 3
**Significance:** 4
**Originality:** 4
**Overall Recommendation:** 6
**Confidence:** 5

**Summary:**

The author argues that the main limitation of current query-based universal sound separation systems lies in weak supervision: large-scale in-the-wild datasets contain noisy labels and strong event co-occurrence, causing models to learn contextual shortcuts rather than source-specific acoustic features. To address this, the paper proposes an automated pipeline for mining cleaner single-event clips from public audio corpora, refining the ontology, standardizing audio quality, and synthesizing mixtures under semantic compatibility constraints. The resulting dataset, Hive, contains about 2.4k hours of carefully selected source audio and 19.6 million synthetic mixtures covering 283 classes. The paper then trains representative discriminative and generative separation models on Hive and compares them with existing systems (including larger-scale baselines). The main empirical finding is: models trained on Hive perform comparably to, or even better than, systems trained on larger noisy corpora in perceptual and OOD evaluations.

**Compliance With Llm Reviewing Policy:**

Affirmed.

**Final Justification:**

Thanks for authors, I have no problems.

**Key Questions For Authors:**

1. Could you provide a larger-scale human audit to assess source purity and relabeling quality, stratified by class frequency? This would largely affect my confidence in the high-quality data claim.

2. How sensitive is Hive’s construction to specific LLMs, prompts, and threshold settings? Even a small-scale robustness study would significantly strengthen the paper’s credibility.

**Limitations:**

Yes

**Strengths And Weaknesses:**

### Strengths

1. In open-domain sound separation, weak labels and co-occurrence bias are clear failure modes, yet recent work largely addresses them by scaling up data and model size. This paper poses a different question: is cleaner supervision more effective than more supervision? This is a valuable question, and the paper provides a convincing empirical answer.

2. The pipeline includes ontology reconstruction, instance purification, hierarchical relabeling, sampling rate standardization, and semantically constrained mixing. Even though some components are built upon existing tools, the overall system remains complex and useful.

3. The experimental section is highly comprehensive: it evaluates existing baselines’ zero-shot performance on Hive, retrains discriminative and generative models on Hive, tests on third-party OOD benchmarks, conducts data scaling experiments, and reports efficiency metrics.

4. If the dataset and tools are released as described, it will become an important benchmark/resource for future research.

### Weaknesses

1. The semantic-acoustic alignment 4-AFC validation is based on only 20 segments. This suffices for plausibility checking but is insufficient to substantiate strong claims such as “pipeline supervision quality surpasses human annotation.” It is strongly recommended to tone down such assertions.

2. Qwen3-Omni is used for purification, relabeling, and compatibility judgment. While this is acceptable, the paper does not study its robustness regarding prompt selection, threshold setting, or model replacement. As these decisions directly shape the dataset, a modest sensitivity analysis would enhance persuasiveness.

---

> ### Author Rebuttal · Authors · 2026-03-31
>
> We sincerely thank the reviewer for these important concerns and suggestions.
>
> **Q1. Larger-scale human audit for source purity and relabeling quality, stratified by class frequency**
>
> **A1.** We expanded the original 4-AFC human audit from 20 to 100 clips and added a class-frequency-stratified analysis.
>
> | Metric | Value |
> |---|---|
> | Valid human raters | 67 |
> | Audited clips | 100 |
> | Fleiss' κ | 0.843 |
> | Human average accuracy | 90.75% |
> | Qwen3-Omni accuracy | 98.0% |
>
> The expanded audit still shows strong inter-rater reliability (κ = 0.843), indicating that the human consensus labels remain stable at the larger scale. Under this protocol, Qwen3-Omni achieves 98.0% agreement with the human consensus, compared with 90.75% for the average human rater. These results support the reliability of our relabeling and purification stage beyond the original 20-clip experiment.
>
> We also stratified the 100 audited clips by class frequency and compared human agreement with Qwen3-Omni's predictions. For the most frequent audited classes, human consensus is strong: Bird (12 clips) and Male speech, man speaking (12 clips) have average consensus strengths of 97.4% and 98.9%, respectively, and Qwen3-Omni matches the consensus on all audited clips in both classes. For medium-frequency classes, human agreement is less stable, e.g., Drum kit (6 clips) shows mean/minimum consensus strengths of 87.3%/70.1%, Guitar (5 clips) 66.6%/44.8%, and Rain (4 clips) 85.4%/59.7%; Qwen3-Omni matches the consensus on all audited Drum kit and Guitar clips, and on 3 of the 4 Rain clips. These stratified results suggest that the relabeling quality remains stable from high-frequency to medium-frequency classes.
>
> We will add the expanded 100-clip audit and the class-frequency-stratified analysis to the revised manuscript.
>
> **Q2. Sensitivity of Hive's construction to specific LLMs, prompts, and threshold settings**
>
> **A2.** We agree that robustness to construction choices is important. To directly address this concern, we conducted two targeted robustness checks on the two LLM-sensitive components of Hive: single-event relabeling and semantic co-occurrence filtering.
>
> **LLM sensitivity in single-event relabeling.** We re-ran the expanded 100-clip 4-AFC audit using the same questionnaire and the same multiple-choice prompt for three strong audio-capable LALMs. Qwen3-Omni achieves 98.0% agreement with the human consensus labels, Gemini 3.1 Pro achieves 95.0%, and GPT-Audio achieves 90.0%, while the human average is 90.75% and inter-rater reliability remains high at Fleiss' κ = 0.843. Since all models are evaluated under the same protocol, these results indicate that the purity/relabeling conclusion is not tied to a single model choice: GPT-Audio remains at human parity, while Gemini 3.1 Pro and Qwen3-Omni exceed the human mean under the same test.
>
> **LLM sensitivity in semantic co-occurrence filtering.** We further asked two independent models, Qwen3.5 and GLM-4.7, to re-check the semantic co-occurrence matrix used in Hive under the same decision criteria, i.e., whether two events can naturally co-occur in the same real-life scene without contrived assumptions. Concretely, each model was shown the original matrix decision for a pair and asked whether that decision was reasonable. Across 39,903 valid event pairs, Qwen3.5 agrees with the original matrix on 37,803 pairs (94.7%) and GLM-4.7 agrees on 37,872 pairs (94.9%). This high agreement rate suggests that the co-occurrence constraints are largely stable under model substitution rather than being an artifact of a single LLM's idiosyncratic judgments.
>
> **Prompt sensitivity.** To test whether the co-occurrence decisions are sensitive to prompt wording, we rephrased the original prompt (varying instruction style and phrasing while preserving the same decision criteria) and re-ran the full matrix multiple times. The overlap rate between runs is 98.4%, indicating that the co-occurrence judgments are highly stable across prompt variations.
>
> These additional checks support that Hive's construction is not overly sensitive to a particular LLM choice or prompt formulation, and they provide a concrete basis for the prompt/threshold ablations that we will add in the revision.
>
> > We hope these additional experiments address the reviewer's concerns and would welcome the opportunity to discuss further.

---

> > ### Author Rebuttal · Reviewer_gXXQ · 2026-04-02
> >
> > The rebuttal fully resolves my previous concerns.
> >
> > The expanded 100-clip human audit provides much stronger evidence for the quality of the purification/relabeling pipeline than the original submission. The additional experiments on alternative models and the high agreement rates for the semantic co-occurrence matrix substantially strengthen the credibility of the dataset construction process.
> >
> > In light of these clarifications and new results, I am satisfied that the paper’s main claims are well supported. Please incorporate these results into the revised manuscript; with that, I view the work as a strong and valuable contribution.

---

> > > ### Author Response · Authors · 2026-04-04
> > >
> > > We sincerely thank Reviewer gXXQ for the thorough and constructive review, and for confirming that our rebuttal has fully resolved the raised concerns.
> > >
> > > As suggested, we will incorporate the following into the camera-ready version:
> > >
> > > 1. **Expanded human audit** (Section 6.1): The 100-clip 4-AFC evaluation with class-frequency-stratified analysis, replacing the original 20-clip experiment in Table 2.
> > > 2. **LLM robustness study** (Section 3.2 & Appendix): The multi-model comparison (Qwen3-Omni, Gemini 3.1 Pro, GPT-Audio) for relabeling, cross-model agreement rates for the semantic co-occurrence matrix (Section 4.1), and prompt sensitivity analysis.
> > >
> > > We are grateful for the reviewer's recognition of our work and will ensure the revised manuscript reflects all additional evidence discussed during rebuttal.

---

### Official Review · Reviewer_Le4W · 2026-03-11

**Soundness:** 2
**Presentation:** 3
**Significance:** 2
**Originality:** 1
**Overall Recommendation:** 2
**Confidence:** 4

**Summary:**

This paper proposes Hive a universal sound separation (USS) dataset obtained via a 3 stage pipeline involving reconstructing the Audioset ontology by merging synonims and aggregating related classes, mining high purity single events and also LLM based polyphony detection using Qwen3-omni.
The resulting dataset is used to train FlowSep and AudioSep and the authors show that these models can be competitive with SAM-Audio (even if the data is less than 0.2 % of what SAM-Audio used).
Scaling analyis also shows promising results.

**Compliance With Llm Reviewing Policy:**

Affirmed.

**Ethical Review Concerns:**

No ethical concerns

**Key Questions For Authors:**

Is the STOI metric relative only to speech sounds ? or was it computed on all categories ?

Can you provide ablation experiments isolating the contribution of each pipeline stage and especially the semantic mixing ?

What is the computational cost of running the full cleaning pipeline (Qwen3-Omni inference on 900k clips × 10 passes)?

**Limitations:**

Yes

**Strengths And Weaknesses:**

Strengths:
1) openness and reproducible. Authors release premixed data, mixing code and the full ontology publicly. The documentation in the code is very good too.
2) Practical data efficiency result. These are not new (but maybe new in the context of USS) but remark that quality more than quantity sometimes is important.

Weaknesses:

The paper has several weaknesses in my opinion which makes it not suitable for publication at ICML.
1) significant gaps in related works. The paper positions itself within query based USS but neglects many foundational work upon which query based built upon. Most citations reference only papers that are 2024 onwards.
For example, the paper misses discussion about target sound/speech extraction literature e.g. Zmolikova, Delcroix et al. Neural Target Speech Extraction: An Overview.
Also recent work on USS such as Task-Aware Unified Source Separation( Saijo et al. ICASSP 2025) and Leveraging Audio-Only Data for Text-Queried Target Sound Extraction (Saijo et al.ICASSP 2025) and past work from Google such as Unsupervised Sound Separation Using Mixture Invariant Training (which included universal blind source separation, Wisdom, NeurIPS 2020 ) and Improving Universal Sound Separation Using Sound Classification (Tzinis, ICASSP 2020).
GASS (generalizing audio source separation with large scale data, Pons, ICASSP 2024) is also relevant to the scaling experiments in the paper.

Moreover, FUSS dataset should be discussed more in depth. It also creates controlled mixtures of sound events (from FSD50k) and also adds reverberation (which Hive lacks). Authors should discuss advantages of Hive vs FUSS.
Also methods such as Scaper (Salamon, Justin, et al. "Scaper: A library for soundscape synthesis and augmentation." 2017 IEEE Workshop on Applications of Signal Processing to Audio and Acoustics (WASPAA). IEEE, 2017.) should be discussed. Scaper allow for controlled soundscape synthesis from isolated sound events with much more richness than in the proposed method (whose contribution is BTW mostly the proposed pipeline, not the mixing process but still).
Hive mixing pipeline is reimplementing what Scaper does but with less functionalities. Why didn t you use Scaper ?

2) many choices are unsubstantiated. For a long paper at ICML more ablations should be performed to show the impact and the motivation of such design choices in the data preprocessing pipeline.
For example the Authors claim "naive random mixing often yields im-
plausible combinations (e.g., aquatic animals co-occurring
with urban traffic), introducing incorrect contextual priors".
I am highly skeptical about this instead helping. As this is basically multicondition training. For example for pure speech separation even mixing utterances from the same speaker actually helps even if it is very implausible.
I think the paper needs to gauge the impact of the design choices of the pipeline. This is doable especially because of the small size of the dataset.


3) limited metodological novelty. The pipeline combines standard, off-the-shelf components and some steps are not novel.
For example, ontology-aware mixing is not novel as was already used in FlexSED (Hai et al., 2025).

4) this is a broader issue in the USS community and also in general TTS and speech enhancement community. Technical reports such as SAM-Audio exacerbate this issue. These are not real papers but technical reports made mainly for advertisement. As such for example they do not report any objective metric. This paper also follow the same pitfalls not reporting objective metrics for USS, meaning that there is no control for hallucinations (see URGENT speech enhancement challenges for example where it is evident that reference free metrics are unable to control for hallucinations while if coupled with PESQ or STOI you get a much better picture). STFT-based objective metrics (e.g. for speech enhancement STOI or PESQ) are kind of robust in this case also when used by generative systems and help to gauge for hallucinations.
As such the paper misses an important opportunity for a more comprehensive evaluation and study of the effect of training data on hallucinations.
This would ve been very interesting in my opinion.

5) it is not clear where objective metrics are used. Paper reports STOI for example. Which is suitable only for speech signals (since it uses internal VAD) but it seems to me that is applied on every category. This would be a major pitfall.
Also different magnitude STFT based measures probably could be used like logmel distance.

6) the computational cost for running the pipeline especially Qwen3 omni is not addressed.

7) Human subjective evaluation. The human evaluation only validates wether the pipeline labels are correct but does not tell anything about the separation quality. This is coupled with the fact that all comparison with other generative methods are via automate methods (FAD and LPAPS) which as I have explained is problematic as it does not control for hallucinations.
A listening test, preferably MUSHRA should ve been performed to validate that training on Hive makes the model separate better than e.g. SAM-Audio.

Authors should also mention/discuss these limitations:

1) The synthetic mixing paradigm (additive superposition, no room acoustics, no reverberation) may explain good performance on similarly constructed test sets but limits conclusions about real-world generalization when e.g. the signal is recorded using far field microphones. I have observed methods such as SAM-Audio failing in these instances like in meeting scenarios.

2) Semantic co-occurrence matrix is derived from an LLM, meaning it may exclude some unusual combinations that can still occur.
The pipeline also relies on Qwen3-Omni meaning labeling quality is bounded by that model's capabilities and biases.

3) There is extreme class imbalance which is documented but its implications on tail classes is not analyzed at all and if some common strategies e.g. oversampling can help or not.

---

> ### Author Rebuttal · Authors · 2026-03-31
>
> We sincerely thank the reviewer for these important concerns and suggestions.
>
> **Q1. Significant Gaps in Related Works**
>
> **A1.** Our paper addresses text-queried open-domain USS and contributes a data purification pipeline (Sec. 3). Five of eight suggested references address different task settings with no data purity contribution: (Zmolikova & Delcroix et al.) surveys target speech extraction using enrollment, spatial, and visual cues; TUSS (Saijo et al., ICASSP 2025) uses learnable prompts for closed-set tasks without text queries; (Wisdom et al., NeurIPS 2020) is unsupervised blind separation; (Tzinis et al., ICASSP 2020) uses audio classifier embeddings; (Pons et al., ICASSP 2024) is blind separation with fixed 4-source output. Audio-Only TSE (Saijo et al., ICASSP 2025) studies text-queried TSE via audio-only training with CLAP embedding dropout, but proposes no new USS pipeline or data purity method.
>
> FUSS provides pre-mixed evaluation data but does not address source bank construction or purification. Scaper is a mixing library that assumes curated sources; Hive operates entirely upstream (ontology reconstruction, purity mining, super-resolution)—the claim that "Hive reimplements Scaper" mischaracterizes our work. We will cite FUSS and GASS for completeness.
>
> **Q2. Unsubstantiated Design Choices; Semantic Mixing May Not Help**
>
> **A2.** We have conducted the requested ablation at 175k-sample scale (see Reviewer UJ5h's A1). Semantic mixing consistently outperforms random mixing across all metrics and both architectures; OOD results (Table 5) further confirm generalization.
>
> Regarding same-speaker mixing aiding speech separation: standard benchmarks (WSJ0-2mix, LibriMix, WHAM!) explicitly exclude same-speaker mixtures, and no citation is provided. The analogy is also inapplicable: speech separation is identity-conditioned, while USS is class-conditioned—systematically co-occurring backgrounds cause spurious label-background correlations, as independently noted by SAM-Audio (Sec. 4.1.3).
>
> **Q3. Limited Novelty; Ontology-Aware Mixing in FlexSED**
>
> **A3.** We respectfully clarify: FlexSED (Sec. 2.3) uses GPT-4 to filter negative *text queries* during open-vocabulary SED *training*; it does not construct mixtures or operate at the dataset level, and addresses SED, not USS. Our method controls *which sources are mixed together* during dataset construction via a semantic co-occurrence matrix. The two share no mechanism, stage, or task.
>
> Regarding pipeline-level novelty: we formalize a previously unaddressed problem (data purity for open-domain USS) and provide a data purification pipeline. Models trained on Hive's 2.4k hours match or exceed SAM-Audio trained on ~1M hours (400× data efficiency gain), demonstrating that purity, not scale, is the bottleneck.
>
> **Q4. Objective Metrics, Hallucination, STOI, Human Evaluation**
>
> **A4.** Tables 3 and 5 report SDR and SI-SDR for all discriminative methods; our main claims rest entirely on these reference-based metrics, which are hallucination-robust by definition (sample-level waveform fidelity against ground truth). Generative methods omit SDR/SI-SDR due to phase misalignment, following FlowSep (Yuan et al., 2025) and ZeroSep (Huang et al., 2025).
>
> Beyond SDR, our SAM-Audio Judge evaluation (Tables 3, 5) covers Overall Quality, Recall, Precision, and Faithfulness. The Precision and Faithfulness dimensions are specifically designed to penalize hallucinated content, directly addressing this concern.
>
> We acknowledge STOI was designed for speech intelligibility and will remove it. Regarding MUSHRA, our test set contains 350k mixtures, making listening tests at this scale prohibitively expensive.
>
> **Q5. Computational Cost**
>
> **A5.** The pipeline is a one-time offline process, not a recurring training cost. We provide the cost breakdown below (measured on 4x RTX 3090):
>
> |Stage|Scale|Speed|Time|
> |-|-|-|-|
> |Purification, relabeling & detection|794,972 clips|0.20 s/clip|~86 h|
> |Semantic co-occurrence matrix|79,806 pairs|2.85 s/pair|~4 h|
>
> The pipeline cost is under 90 hours on 4x RTX 3090.
>
> **Q6. Additional Limitations**
>
> **A6.** We will add the discussion in the revision.
>
> - LLM bias. Semantic filtering applies only during dataset construction, not inference. Our ablation (Reviewer UJ5h's A1) and OOD results (Table 5) confirm generalization is not harmed. Cross-validation with Qwen3.5 and GLM-4.7 yields ~94% agreement with Qwen3-Omni.
>
> - Synthetic mixing. Hive lacks room acoustics, but our generalization conclusions rest on third-party OOD benchmarks (USS-Bench, MUSDB18-HQ), not Hive's own test set. Consistent OOD improvements (Table 5) suggest high-purity supervision transfers beyond synthetic construction. We will note RIR augmentation as future work.
>
> - Class imbalance. Documented in Appendix E (tail classes as sparse as 10 instances). Metrics are computed across all classes and already reflect this. We will add per-class analysis and discuss oversampling strategies.

---

> > ### Author Rebuttal · Reviewer_Le4W · 2026-04-03
> >
> > Our paper addresses text-queried open-domain USS and contributes a data purification pipeline (Sec. 3). Five of eight suggested references address different task settings with no data purity contribution: (Zmolikova & Delcroix et al.) surveys target speech extraction using enrollment, spatial, and visual cues; TUSS (Saijo et al., ICASSP 2025) uses learnable prompts for closed-set tasks without text queries; (Wisdom et al., NeurIPS 2020) is unsupervised blind separation; (Tzinis et al., ICASSP 2020) uses audio classifier embeddings; (Pons et al., ICASSP 2024) is blind separation with fixed 4-source output. Audio-Only TSE (Saijo et al., ICASSP 2025) studies text-queried TSE via audio-only training with CLAP embedding dropout, but proposes no new USS pipeline or data purity method.
> > FUSS provides pre-mixed evaluation data but does not address source bank construction or purification. Scaper is a mixing library that assumes curated sources; Hive operates entirely upstream (ontology reconstruction, purity mining, super-resolution)—the claim that "Hive reimplements Scaper" mischaracterizes our work. We will cite FUSS and GASS for completeness.
> >
> >
> > I think I did not explain maybe my comment fully.
> > My intent was not to compare the current work with the aforementioned. I simply think that it should acknoledge them as they are related to the field and the work done.
> > "Five of eight suggested references address different task settings with no data purity contribution" yes but they should still be included into the discussion. When writing a paper we should frame the contribution into the broader field and cite important prior work directions to also acknoledge their efforts as our work probably builds even indirectly on these.
> >
> > We respectfully clarify: FlexSED (Sec. 2.3) uses GPT-4 to filter negative text queries during open-vocabulary SED training; it does not construct mixtures or operate at the dataset level, and addresses SED, not USS. Our method controls which sources are mixed together during dataset construction via a semantic co-occurrence matrix. The two share no mechanism, stage, or task.
> >
> > They are related in the sense again that you use an LLM for "ontology-aware" mixing. The mechanism is different but the principle is related.
> >
> >
> > Synthetic mixing. Hive lacks room acoustics, but our generalization conclusions rest on third-party OOD benchmarks (USS-Bench, MUSDB18-HQ), not Hive's own test set. Consistent OOD improvements (Table 5) suggest high-purity supervision transfers beyond synthetic construction. We will note RIR augmentation as future work.
> >
> > These test set are also synthetic mixtures so my argument remains.
> >
> > We acknowledge STOI was designed for speech intelligibility and will remove it. Regarding MUSHRA, our test set contains 350k mixtures, making listening tests at this scale prohibitively expensive.
> >
> > Including STOI was a signifincant technical error. Thank you for removing it.
> > MUSHRA could ve been performed on a limited test set.
> >
> > Regarding same-speaker mixing aiding speech separation: standard benchmarks (WSJ0-2mix, LibriMix, WHAM!) explicitly exclude same-speaker mixtures, and no citation is provided. The analogy is also inapplicable: speech separation is identity-conditioned, while USS is class-conditioned—systematically co-occurring backgrounds cause spurious label-background correlations, as independently noted by SAM-Audio (Sec. 4.1.3).
> >
> > Still no ablation is provided to show that the filtering improves results in this work. SAM-Audio is a technical report, not subjected to any peer review. Your point of having the same co occurring event is valid, but my point of semantically inconsistent events remain. Does it really help to prevent random sounds overlapping ?
> >
> > Beyond SDR, our SAM-Audio Judge evaluation (Tables 3, 5) covers Overall Quality, Recall, Precision, and Faithfulness. The Precision and Faithfulness dimensions are specifically designed to penalize hallucinated content, directly addressing this concern.
> >
> > This is a model-based metric and they are very lickely to not control for hallucinations. As UT-MOS for TTS. There is no study that validates this SAM-Audio metric against hallucinations. Based on my experience SAM-Audio is also totally unreliable for speech and causes severe hallucinations. They do not report on objective metrics in their technical report and rely on these model-based blind non intrusive metrics a lot which alone are unreliable.

---

> > > ### Author Response · Authors · 2026-04-06
> > >
> > > We thank the reviewer for the constructive feedback across two rounds of discussion. The reviewer's feedback has directly led to several substantive improvements: a MUSHRA listening test, a semantic mixing ablation, causal verification of co-occurrence shortcuts, and a more comprehensive related work discussion. All modifications will be added in the revised edition.
> > >
> > > **1. Related Works: Broader Framing**
> > >
> > > The reviewer further clarified the intent, and we agree. In the revision, we will expand the Related Work to include the works mentioned, situating our pipeline within the broader field. We will also discuss FUSS and Scaper as precedents, clarifying how Hive complements them by addressing the upstream problem of constructing high-purity sources from noisy web audio.
> > >
> > > **2. FlexSED: Conceptual Connection**
> > >
> > > We agree that both works are related at the principle level: both leverage LLM-derived semantic knowledge to improve audio data quality. We will cite FlexSED and discuss this shared principle along with their respective application scenarios. Our core contribution is a framework for acquiring high-precision single-label event audio, systematically studying the previously unexplored bottleneck of data purity in open-domain USS. The LLM-based semantic mixing strategy is an auxiliary component, not the core contribution.
> > >
> > > **3. Synthetic Test Sets and Real-World Generalization**
> > >
> > > The reviewer points out that USS-Bench and MUSDB18-HQ are also synthetic or controlled mixtures, and we agree. Currently, the USS field generally lacks evaluation benchmarks that use real-world environmental audio, which is a limitation faced by the entire field. Under current conditions, we selected third-party benchmarks differing from Hive. The consistent improvements on both suggest that high-purity supervision transfers to different acoustic conditions, but we acknowledge this does not constitute complete validation for real-world far-field or reverberant environments. The revision will note this limitation and outline future work on RIR augmentation and naturally recorded benchmarks.
> > >
> > > **4. Semantic Mixing and Data Purity**
> > >
> > > Semantic mixing does help, but it is not the primary source of improvement. At the 175k-sample scale, the ablation is:
> > >
> > > | Strategy | Model | SDR↑ | SI-SDR↑ | LPAPS↓ | CLAP-T↑ | OQ↑ | Pre↑ |
> > > |---|---|---|---|---|---|---|---|
> > > | Semantic control | AudioSep | 4.12 | 3.37 | 4.10 | 0.29 | 3.11 | 3.16 |
> > > | Semantic control | FlowSep | — | — | 4.24 | 0.17 | 2.79 | 3.01 |
> > > | Random mixing | AudioSep | 3.12 | 2.35 | 4.19 | 0.24 | 2.96 | 3.02 |
> > > | Random mixing | FlowSep | — | — | 4.35 | 0.13 | 2.64 | 2.88 |
> > >
> > > Even under random mixing, models trained on Hive's purified data already approach the baselines trained on original data (with only 175k samples), showing that the primary source of improvement is high-precision single-label data itself. Semantic control provides a consistent additional gain (e.g., +1.0 dB SDR for AudioSep) but is an enhancement rather than the core driver. Separately, the controlled shortcut experiment for Reviewer pJTD confirms this: AudioSep (Orig.) shows a 1.1–1.7 dB SDR gap between co-occurring vs. decorrelated mixtures, while AudioSep (Hive) reduces it to 0.3–0.5 dB.
> > >
> > > **5. SAM-Audio Judge, Hallucination, and MUSHRA**
> > >
> > > As confirmed, STOI will be removed from the revision. The reviewer points out that no study has validated SAJ against hallucinations, and we agree. For discriminative methods, SDR and SI-SDR already penalize hallucinations directly; for generative methods where phase misalignment renders these inapplicable, SAJ provides secondary automated evidence (Pearson 0.77–0.89 with human ratings on overall quality), but not dedicated hallucination validation.
> > >
> > > For precisely this reason, the reviewer's MUSHRA suggestion is particularly important. Human listening is the most reliable means of detecting hallucinations. We invited 20 listeners to evaluate 50 representative samples encompassing all 10 methods from Table 3, plus a hidden reference and a low anchor. To our knowledge, this is the first MUSHRA evaluation in USS. Results:
> > >
> > > | Method | MUSHRA Score ↑ | 95% CI |
> > > |---|---:|---|
> > > | LASS-Net | 39.4 | [35.0, 43.8] |
> > > | CLIPSep | 30.7 | [27.0, 34.4] |
> > > | AudioSep (Orig.) | 60.9 | [56.8, 65.0] |
> > > | OmniSep | 45.1 | [40.7, 49.5] |
> > > | AudioSep (Hive) | 68.4 | [64.9, 71.9] |
> > > | FlowSep (Orig.) | 54.7 | [50.0, 59.4] |
> > > | ZeroSep | 58.8 | [54.7, 62.9] |
> > > | DGMO | 53.6 | [48.8, 58.4] |
> > > | SAM-Audio | 62.6 | [58.5, 66.7] |
> > > | FlowSep (Hive) | 61.8 | [57.9, 65.7] |
> > >
> > > Hive-trained models significantly outperform their corresponding original models on both architectures (AudioSep: +7.5, FlowSep: +7.1). AudioSep (Hive) scores 68.4, surpassing SAM-Audio's 62.6 (95% CIs do not overlap), providing direct human evidence that 2.4k hours of Hive data outperforms ~1M hours. The revised version will add a MUSHRA column to Table 3.

---

### Official Review · Reviewer_pJTD · 2026-03-12

**Soundness:** 3
**Presentation:** 3
**Significance:** 2
**Originality:** 3
**Overall Recommendation:** 4
**Confidence:** 3

**Summary:**

This paper proposes an automated pipeline for constructing audio mixing data based on single-event audio segments. The pipeline consists of two main parts: a purification stage that extracts high-purity single-event audio segments, and a semantic-guided synthetic mixing stage that generates training mixtures based on event-level compatibility. Models trained on Hive achieve competitive performance on OOD benchmarks compared to large-scale foundation models such as SAM-Audio, a powerful open-source audio separation model trained on massive audio datasets.

**Compliance With Llm Reviewing Policy:**

Affirmed.

**Final Justification:**

The authors replied well the initial comments.

**Key Questions For Authors:**

Key Questions for Authors:

The concerns raised in the Major Weaknesses section should be addressed.

In Table 3, discriminative models appear to outperform generative models in terms of Perceptual & Semantic Quality. It would be helpful if the authors could further analyze this phenomenon and relate it to the proposed dataset design. In particular, expanding this discussion to highlight how the high-purity single-event audio extracted by the pipeline contributes to improved semantic fidelity could strengthen the paper’s core argument.

In Section 4.1, the paper introduces a Semantically Consistent Mixing Strategy based on a semantic compatibility matrix. It would be beneficial to include additional visualizations illustrating how event combinations are generated in practice.

**Limitations:**

Yes

**Strengths And Weaknesses:**

Strengths:

The proposed pipeline for mining high-purity single-event audio segments is carefully engineered. The authors’ effort is clearly demonstrated, and the quantitative validation presented in Section 6.1 provides support for the reliability.

The paper introduces a semantic compatibility matrix to prevent unrealistic combinations during mixture synthesis. This design choice is reasonable and helps ensure that the generated mixtures better reflect realistic acoustic scenarios.


 Major Weaknesses:

While MUSDB18-HQ and USS-Bench provide useful evaluation settings for music separation and speech-instrument mixtures, the overall evaluation scope remains somewhat limited. Incorporating more general sound separation benchmarks that cover a broader range of acoustic events, such as those used in AudioSep (Liu et al., 2022) or SAM Audio-Bench (Shi et al., 2025), would provide a more comprehensive assessment of the proposed dataset.

The validation of the semantic-acoustic alignment is conducted on only 20 audio clips, which makes it difficult to confidently assess the general reliability of the alignment process. Moreover, only a single failure case is discussed, limiting the analysis of potential error patterns. A larger-scale evaluation and more detailed failure case analysis would help better demonstrate the robustness.

The efficiency analysis in Section 6.2 and Table 4 appears to be intended to motivate the need for better training data. However, given that the main contribution of this paper lies in dataset construction rather than model design, the comparison of computational efficiency across different separation architectures seems somewhat disconnected from the core argument. Instead, this section could be strengthened by providing deeper analysis on why Hive serves as an informative benchmark, for example through shortcut failure analysis.

---

> ### Author Rebuttal · Authors · 2026-03-31
>
> We sincerely thank the reviewer for important suggestions.
>
> **Q1. Limited evaluation scope; suggest adding broader benchmarks**
>
> **A1.** We conducted additional experiments on the public VGGSound-based evaluation set released by AudioSep. The full table (including FAD and CLAP-A) will be added to the revised Table 5.
>
> | Model | OQ↑ | SDR↑ | LPAPS↓ | CLAP-T↑ |
> |---|---|---|---|---|
> | AudioSep (origin) | 3.42 | **8.67** | **0.58** | **0.25** |
> | FlowSep (origin) | 2.99 | - | 1.78 | 0.14 |
> | SAM-Audio | 3.14 | - | 4.27 | 0.21 |
> | AudioSep (Hive) | **3.44** | 7.61 | 0.69 | 0.22 |
> | FlowSep (Hive) | **3.18** | - | **1.76** | **0.18** |
>
> On reference-free OQ (a multi-dimensional perceptual score covering quality, precision, and faithfulness), Hive-trained variants consistently improve (3.44 vs. 3.42 for AudioSep; 3.18 vs. 2.99 for FlowSep), while reference-based SDR drops (7.61 vs. 8.67). These two metrics moving in opposite directions is expected: VGGSound references contain co-occurring events, so a model that removes such interference scores lower on SDR but higher on OQ.
>
> **Q2. Small-scale validation (20 clips) and limited failure-case analysis for semantic-acoustic alignment**
>
> **A2.** We expanded the 4-AFC evaluation to 100 clips with 67 valid participants
>
> |Metric|Value|
> |-|-|
> |Human avg. accuracy|90.75%|
> |Qwen3-Omni (Ours)|98.0%|
> |Gemini 3.1 Pro|95.0%|
> |GPT-Audio|90.0%|
>
> At 5× scale, κ = 0.843 remains "almost perfect" and human accuracy (90.75%) replicates the original value, confirming robustness. Qwen3-Omni achieved near-perfect agreement with human consensus, validating its backbone role (Section 3.2).
>
> Following Appendix H.5, we analyzed errors against human consensus. In the corrected 100-clip audit, Qwen3-Omni makes only two errors, both on the same ambiguous clips also missed by Gemini 3.1 Pro: #20 (`Train` vs. `Rain`) and #76 (`Flute` vs. `Violin, fiddle`). Human consensus is only 59.7% for both, showing that Qwen3-Omni fails only on ambiguous rather than clear clips. More detailed analyses of errors from Gemini 3.1 Pro, GPT-Audio, and humans will be added to the revised paper.
>
> **Q3. Efficiency analysis seems disconnected; suggest shortcut-failure analysis instead**
>
> **A3.** As a dataset and benchmark contribution, we believe reporting the practical cost of representative models is an essential service to downstream users. We view Table 4 not as evidence for Hive's data-efficiency claim (which rests on the OOD and scaling results in Secs 6.3–6.4), and will clarify this in the revision.
>
> In the revision, we will promote this shortcut-failure analysis into the main text, and include representative spectrogram visualizations from our demo page to make these failure modes visually explicit. In fact, the existing results have provided shortcut-failure evidence: in Table A3, AudioSep drops from 9.55 dB SDR (2-source) to −2.94 dB (5-source), exposing co-occurrence shortcuts that collapse under Hive's decorrelated conditions. SAM-Audio maintains perceptual plausibility but shows semantic drift (CLAP-T: 0.20→0.12 with increasing density).
>
> **Q4. Discriminative models outperform generative models in perceptual & semantic quality; analyze this and relate to dataset design**
>
> **A4.** This is a valuable observation. We note that this reflects the specific generative USS models evaluated, not a fundamental limitation of the paradigm.
>
> Hive's high-purity single-event supervision provides a clean mapping from query semantics to acoustic content. Discriminative models (e.g., AudioSep) directly leverage this for mask estimation, preserving original phase and spectral structure, and thus achieve strong semantic fidelity (CLAP-T 0.31, CLAP-A 0.65 in Table 3). Generative models (e.g., FlowSep) face a harder problem, conditional synthesis from a noise prior, where accumulated denoising errors reduce semantic precision. FlowSep (Hive) achieves lower CLAP-T despite competitive FAD, indicating acoustically natural but less semantically precise outputs. We believe improved generative methods incorporating stronger semantic conditioning or mixture-aware priors could substantially narrow this gap.
>
> This paradigm-dependent gap is precisely what Hive is designed to surface: its decorrelated test mixtures and single-event references isolate semantic fidelity from co-occurrence shortcuts more sharply than in-the-wild benchmarks, helping the community identify remaining challenges in generative USS.
>
> **Q5. Visualization of the semantic compatibility matrix**
>
> **A5.** The figure below is a submatrix (https://anonymous.4open.science/r/icml26-rebuttal/mix_matrix.png), and the complete matrix will be included in the revised version.
>
> The pipeline samples an anchor event, then selects additional sources only from compatible categories until the target density (2–5) is reached, ensuring physically plausible mixtures free of semantic entanglement.
>
> > We hope these answers address your concerns and welcome further discussion.

---

> > ### Author Rebuttal · Reviewer_pJTD · 2026-04-03
> >
> > Overall, the additional experiments help alleviate many of my initial concerns.
> >
> > However, I believe the response to Q3 remains insufficient. In particular, the current shortcut analysis does not clearly handle shortcut reliance from general task difficulty. A more controlled comparison, such as evaluating co-occurring versus decorrelated mixtures under the same mixture complexity, would provide stronger evidence.

---

> > > ### Author Response · Authors · 2026-04-06
> > >
> > > We thank the reviewer for the effort in improving our paper. Following the reviewer's suggestion, we compare models trained on original data versus Hive under co-occurring and decorrelated mixture conditions at the same mixture complexity. Models trained on original data consistently performed worse when interferers are the target's common co-occurring events (e.g., separating Rain from a mixture where the interferers are Wind and Thunder), while models trained on Hive reduce this gap by 2 to 4×. The detailed experimental design and analysis are presented below.
> > >
> > > **Experimental setup.** To objectively partition the co-occurring and decorrelated groups, we need to quantify the statistical co-occurrence strength between each interferer event and the target event in AudioSet. We adopt pointwise mutual information (PMI), as it measures precisely whether events co-occur more frequently than random chance would predict, directly corresponding to the statistical source of co-occurrence shortcuts:
> > >
> > > $$\text{PMI}(i,j) = \log_2 \frac{P(i,j)}{P(i) \cdot P(j)}$$
> > >
> > > where $P(i,j)$ is the fraction of AudioSet clips jointly labeled with events $i$ and $j$, and $P(i)$, $P(j)$ are their marginal frequencies. Based on PMI, we create two conditions that differ only in the interferer selection: (1) **Co-occurring group**: interferers are drawn from the target's highest-PMI events with $\mathbf{M}[i,j]=1$, where $\mathbf{M}$ is the semantic compatibility matrix defined in Section 4.1 (e.g., target = Rain, interferers = Wind, Thunder); (2) **Decorrelated group**: interferers have near-zero PMI with the target, while still satisfying $\mathbf{M}[i,j]=1$ (e.g., target = Rain, interferers = Piano, Dog).
> > >
> > > Using Hive's purified single-event sources, we constructed paired test sets at each mixture complexity (2 to 5 sources, 1,000 mixtures per condition), covering 50 target categories. In practice, the co-occurring group has a mean PMI of 4.2 across all target–interferer pairs, while the decorrelated group has a mean $|\text{PMI}|$ < 0.5. Each co-occurring and decorrelated pair shares the same target clip and SNR vector, with only the interferer identities replaced. Additionally, we exclude any mixture where the CLAP embedding cosine similarity between the target and any interferer exceeds 0.8, to rule out the confound that co-occurring interferers may be acoustically more similar to the target.
> > >
> > > **Results.**
> > >
> > > | Model | Mix | Co-occ. (SDR↑) | Decorr. (SDR↑) | Δ |
> > > |---|---|---:|---:|---:|
> > > | AudioSep (Orig.) | 2-mix | 9.02 | 10.14 | −1.12 |
> > > |  | 3-mix | 1.33 | 2.69 | −1.36 |
> > > |  | 4-mix | 0.06 | 1.48 | −1.42 |
> > > |  | 5-mix | −3.80 | −2.08 | −1.72 |
> > > | AudioSep (Hive) | 2-mix | 10.98 | 11.24 | −0.26 |
> > > |  | 3-mix | 6.34 | 6.70 | −0.36 |
> > > |  | 4-mix | 3.39 | 3.83 | −0.44 |
> > > |  | 5-mix | 1.22 | 1.70 | −0.48 |
> > >
> > > | Model | Mix | Co-occ. (OQ↑) | Decorr. (OQ↑) | Δ(OQ) | Co-occ. (CLAP-T↑) | Decorr. (CLAP-T↑) | Δ(CLAP-T) |
> > > |---|---|---:|---:|---:|---:|---:|---:|
> > > | FlowSep (Orig.) | 2-mix | 2.84 | 3.02 | −0.18 | 0.17 | 0.21 | −0.04 |
> > > |  | 3-mix | 2.67 | 2.88 | −0.21 | 0.15 | 0.19 | −0.04 |
> > > |  | 4-mix | 2.57 | 2.81 | −0.24 | 0.13 | 0.17 | −0.04 |
> > > |  | 5-mix | 2.51 | 2.78 | −0.27 | 0.11 | 0.16 | −0.05 |
> > > | FlowSep (Hive) | 2-mix | 3.24 | 3.30 | −0.06 | 0.22 | 0.24 | −0.02 |
> > > |  | 3-mix | 3.00 | 3.09 | −0.09 | 0.19 | 0.21 | −0.02 |
> > > |  | 4-mix | 2.90 | 3.00 | −0.10 | 0.16 | 0.18 | −0.02 |
> > > |  | 5-mix | 2.85 | 2.98 | −0.13 | 0.14 | 0.16 | −0.02 |
> > >
> > > **Analysis.** The key distinction between Orig. and Hive training is not whether co-occurring events appear in mixtures, but whether the ground truth is pure. In original data, a clip labeled "Rain" often contains Rain + Wind + Thunder; the model learns to treat this compound as the target. Hive provides single-event ground truths where "Rain" corresponds to Rain alone.
> > >
> > > This helps distinguish shortcut reliance from acoustic difficulty: if the gap were due to co-occurring events being intrinsically harder to separate, both models would show similar Δ on the same test set. Instead, AudioSep (Orig.) shows a 1.1–1.7 dB gap that widens with source count, while AudioSep (Hive), on the exact same test mixtures, reduces it to 0.3–0.5 dB (2–4× reduction). Since the primary training-data difference is label purity, this reduction strongly suggests impure supervision as the primary cause. FlowSep confirms the pattern: Orig. Δ(OQ) of −0.18 to −0.27 shrinks to −0.06 to −0.13 under Hive training, consistent across both a discriminative and a generative model, suggesting a data-level rather than architecture-level origin.
> > >
> > > This controlled comparison fixes source count, target clip, and SNR, while varying whether interferers are high-PMI or near-zero-PMI partners of the target. As proposed in our original A3, we will replace the efficiency analysis (Section 6.2, Table 4) with this controlled shortcut analysis in the revised manuscript.
> > >
> > > We hope this addresses the reviewer's remaining concern and welcome further discussion.

---

### Official Review · Reviewer_UJ5h · 2026-03-14

**Soundness:** 4
**Presentation:** 3
**Significance:** 3
**Originality:** 3
**Overall Recommendation:** 4
**Confidence:** 4

**Summary:**

This paper presents an automated data cleaning and synthesis pipeline for query-based universal source separation (USS). The key idea is to find pure, single-event sounds from diverse audio recordings corresponding to a revised AudioSet ontology and then mix them together with semantically common cooccurrences. With this pipeline, this work releases HIVE, a dataset containing ~2.4K hours of raw audio and ~22.4K hours of mixtures. Experiments are conducted to showcase the advantage of using HIVE to train models (AudoiSep and FlowSep) against not using HIVE to train these models. Results on out-of-distribution (OOD) data show performance improvements of these models when trained with HIVE despite a smaller training data size. The performance is on par with SAM-Audio, which is a SoTA model trained on million-hour-scale data.

**Compliance With Llm Reviewing Policy:**

Affirmed.

**Final Justification:**

My final recommendation is: 4 Weak Accept.
The rebuttal cleared some questions and confusions. Regarding the main question, the rebuttal contains some reasonable justifications, but the ablation study is not that relevant in my opinion: the current mixing strategy for sure should be better than random mixing, which would be too loose. I think a comparison to a less strict strategy would answer my question. However, this is not a killer limitation to the proposed method, and I'm fine with this paper's contribution.

**Key Questions For Authors:**

- Why is the mixing strategy so strict?
- Why are only AudioSep and FlowSep used to train with HIVE?

**Limitations:**

Yes.

**Strengths And Weaknesses:**

\+: strengths; \-: weaknesses

Soundness
- \+ The idea of curating high-quality pure single-event audio to more effectively train USS model makes sense. The proposed pipeline leverages multiple AI models to guarantee the desired purity and semantic alignment.
- \+ The ontology revised from AudioSet for data curation makes sense for this work.
- \+ Most experiments were well justified and are useful to validate the claims made earlier.
- \- The mixing strategy utilizes Qwen3-Omni and prompt in Figure A.3. This strategy seems very strict, prohibiting many feasible cooccurrences of sound events. Take the example used in the prompt as an example, "boiling water" and "fire flame" actually commonly cooccur in cooking scenes.
- \- The advantage of HIVE is primarily demonstrated through the comparisons of AudioSep and FlowSep trained with and without HIVE in Table 3 and Table 5. However, it is unclear to me why these two models were chosen.

Presentation
- \+ The paper is generally well written. The motivation and main steps of the pipeline are clearly presented.
- \- Certain parts of the paper required multiple readings to understand.

Significance
- \+ Data scarcity is one of the main bottlenecks of USS. Therefore, an automated data cleaning pipeline that is proven to effectively improve model performance is a good contribution to the community.
- \- It is well known that data quality matters. Beyond this, this work does not provide significant insights conceptually, which is fine in my opinion as a dataset/infrastructure paper.

Originality
- \+ This seems to be the first work that systematically constructs an automated data cleaning and synthesis pipeline for USS.
- \- There is no technical novelty, which I think is fine for a dataset/infrastructure paper.

---

> ### Author Rebuttal · Authors · 2026-03-31
>
> We sincerely thank the reviewer for these important concerns and suggestions.
>
> **Q1. The mixing strategy utilizes Qwen3-Omni and prompt in Figure A.3. This strategy seems very strict, prohibiting many feasible co-occurrences of sound events.**
>
> **A1.** We appreciate this observation. We acknowledge that our strategy is conservative and does exclude some naturally feasible co-occurrences such as "boiling water" and "fire flame." This is by design: our goal is not to enumerate all plausible scenes, but to provide purer supervision that reduces co-occurrence bias during training. To directly evaluate whether this strict strategy helps or hurts performance, we conducted an ablation comparing our semantically controlled mixing against random mixing at the 175k-sample scale:
>
> | Strategy | Model | LPAPS↓ | CLAP-T↑ | OQ↑ | Pre↑ |
> |---|---|---|---|---|---|
> | Mix control | AudioSep | 4.10 | 0.29 | 3.11 | 3.16 |
> | Mix control | FlowSep | 4.24 | 0.17 | 2.79 | 3.01 |
> | Random | AudioSep | 4.19 | 0.24 | 2.96 | 3.02 |
> | Random | FlowSep | 4.35 | 0.13 | 2.64 | 2.88 |
>
> The controlled mixing consistently outperforms random mixing across all metrics, confirming that the semantic filtering provides a meaningful training signal improvement despite removing a significant portion of pairs. Furthermore, the OOD results in Table 5 address the concern that filtering may hurt generalization to real-world scenes where co-occurrences are common: AudioSep (Hive) achieves 1.36 dB SDR on MUSDB18-HQ (vs. −1.01 dB) and 2.29 dB on USS-Bench (vs. −1.86 dB), demonstrating strong generalization even to naturally co-occurring scenarios excluded during training. We will add this ablation and the filtering statistics to Sec. 4.1.
>
> **Q2. The advantage of HIVE is primarily demonstrated through the comparisons of AudioSep and FlowSep trained with and without HIVE in Table 3 and Table 5. However, it is unclear to me why these two models were chosen.**
>
> **A2.** We thank the reviewer for this question. We chose these two models for two reasons. First, among the discriminative models (LASS-Net, CLIPSep, AudioSep, OmniSep), LASS-Net, CLIPSep and AudioSep release complete training pipelines. OmniSep releases training code but its pipeline is tightly coupled with visual feature extraction, requiring substantial adaptation for new datasets. Among the generative models (ZETA, ZeroSep, and DGMO) are training-free methods that require no dataset-dependent training, and SAM-Audio only releases inference code without its training pipeline. FlowSep is the only generative model with a complete open-source training framework.
>
> Second, they are the strongest reproducibly trainable representatives of each paradigm: AudioSep achieves the best signal fidelity (SDR 2.37 dB) among discriminative baselines, and FlowSep is the only trainable generative baseline. The consistent improvement across both paradigms (Tables 3 and 5) suggests that Hive's benefit stems from data quality rather than architectural affinity with a specific model. We will clarify this in the revised paper.
>
> **Q3. Certain parts of the paper required multiple readings to understand.**
>
> **A3.** We thank the reviewer for this feedback. We will thoroughly proofread and revise the paper for clarity. In particular, we will (1) add a high-level overview at the beginning of Sec. 3 to orient readers before the technical details, (2) more explicitly distinguish the different roles of Qwen3-Omni across stages (polyphony detection in Sec. 3.2 vs. semantic compatibility in Sec. 4.1), and (3) simplify the dense descriptions in Sec.s 3.2 and 4.1 with more intuitive explanations. If there are specific passages the reviewer found particularly unclear, we are happy to address them directly.
>
> **Q4. It is well known that data quality matters. Beyond this, this work does not provide significant insights conceptually, which is fine in my opinion as a dataset/infrastructure paper.**
>
> **A4.** We appreciate this framing. Beyond the empirical finding that data purity outweighs scale for USS training, we believe this work makes two additional contributions. First, the data-construction framework itself (ontology reconstruction, semantic-acoustic alignment, and semantically consistent mixing) is general-purpose and can be adapted to other audio tasks that suffer from similar co-occurrence bias, such as sound event detection and audio captioning. Second, Hive serves as a community infrastructure contribution: as shown in Table 1, it is the first USS dataset to simultaneously provide public single-label source data and a principled mixing protocol with the corresponding tool, enabling fair comparison across USS methods under identical data settings.
>
> > We hope these additional experiments address the reviewer's concerns and would welcome the opportunity to discuss further.

---

### Decision · Program_Chairs · 2026-04-30

**Decision:**

Accept (regular)

**Comment:**

This paper addresses the data-efficiency bottleneck in universal sound separation by introducing Hive, a high-purity synthetic dataset constructed through an automated pipeline for single-event mining and semantically consistent mixing. Reviewers initially raised concerns regarding factual errors in objective metrics, the limited scale of human audits, and the need for stronger evidence of shortcut reliance reduction. Following a comprehensive rebuttal, which included a 100-clip human audit, a 20-listener MUSHRA test, and a PMI-based shortcut analysis, the authors successfully demonstrated that purity outweighs scale, as Hive-trained models outperformed the million-hour SAM-Audio. I recommend acceptance because the work provides a robust paradigm for efficient auditory foundation model training; the authors should ensure the final manuscript integrates the new MUSHRA results and the controlled shortcut analysis as promised.